# The Alcázar of Córdoba: The Seat of Islamic Power in Al-Andalus

Alberto León-Muñoz 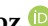

Department of Art History, Archeology and Music, Faculty of Philosophy and Letters, Universidad de Córdoba, 1400 Córdoba, Spain; aa2lemua@uco.es

**Abstract:** In this paper, we show a synthesis of the recovered information in the most recent archaeological interventions of the occupied space by the architectural complex where the Omayyad seat of power and the following leaders of Córdoba were installed. As the most relevant aspects, we show the persistent continuity of the reoccupation and appropriation of the precedent buildings, the tight correlation with the Aljama Mosque, and the architectonic entity of the documented structures.

**Keywords:** Alcázar; Córdoba; Al-Andalus; evolution; Islamic archaeology; urbanism

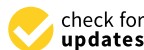



## 1. Introduction

The palace of the Umayyad dynasty, built in the south-western corner of the *medina* of Córdoba after the Islamic conquest of the city, has been a building recurrently mentioned in Arabian chronicle sources. Thanks to these written references, we know of its location, its approximate size, the name of its gates, some of the pavilions that form it, and many of the transcendental events that occurred there or behind its walls. However, the material information of this architectural complex is still relatively unknown, unlike the case of another great Umayyad building, the Congregational Mosque to which it was attached, which underwent successive well-known expansions. While the Islamic oratory has preserved a good part of its architectural integrity, with the exception of the construction of the cathedral in the 16th century, the palace complex was broken down and separated into pieces that were assigned by the Castilian monarchs to the conquest's collaborators (cfr. Escobar Camacho 1989, pp. 127–28; Escobar Camacho 2020, p. 389 ss.), so that the unity of this extensive area was separated into many properties that have undergone an uneven urban evolution, the result being the loss of the complex's image. This lack of a more evident materiality explains its scarce prominence in publications on Umayyad art and architecture.

Another factor to bear in mind is the fact that the urban nature of the area where the Al-Andalus Alcázar is located has undergone far fewer changes than other areas of the city, in which property speculation and building renewal have together led to the completion of various archaeological interventions (Figure 1). The ownership of the Church of an important part of the space occupied by the Alcázar, with buildings identified as having high heritage value (The Episcopal Palace, Seminary of Saint Pelagius, Hospital of Saint Sebastian, etc.), and the transformation of the centre of the palace complex into a large public square (Plaza Campo Santo de los Mártires) have reduced the possibilities of performing preventive archaeological activities until recently. The only exceptions up to the end of the 20th were the discovery and conversion into a museum of the so-called "caliphal baths" in 1903 and in the 1960s, and the activities in the "Bishop's Gardens" at the beginning of the 1970s (*vid. Infra*).

In addition to the above, other factors have conditioned research on the Al-Andalus architectural complex: on the one hand, the difficulty in identifying material evidence recovered from this space, and on the other, the approach by studies in the framework of a philological tradition that has defined decades of work on the Al-Andalus Umayyad capital (*cfr.* León-Muñoz 2022, p. 26). This trend has entailed a certain burden for the

research, which has restricted the possibilities of archaeological analysis, subject to the documentary information.

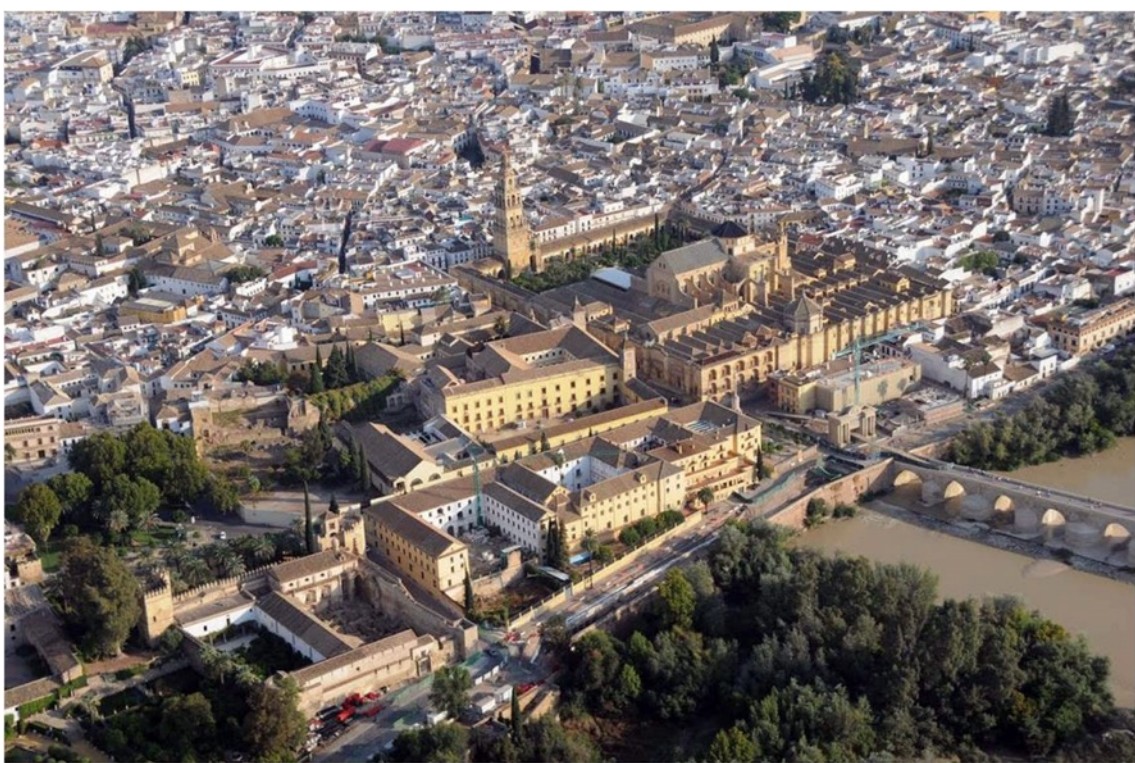

**Figure 1.** Current aerial view of the complex composed of the Mosque-Cathedral and the space occupied by the Alcázar Andalusí (Andalusian Fortress) (image: GMU-UCO Agreement).

Fortunately, in recent years, various archaeological interventions integrated in heritage recovery projects in the south-western sector of the city have provided substantial information regarding the architectural origin and evolution of the Alcázar and have allowed for the approach of a new reading of the space occupied by the seat of Al-Andalus power in Córdoba. It is increasingly evident that the way of looking influences our view of things. Accordingly, the change of paradigm and the observation from the optics of archaeology are transforming the means of conceiving this extensive architectural complex towards a mainly material reading with a diachronic suggestion.

## 2. Previous Research

The status as the seat of Umayyad power in Al-Andalus has made this enclave the focus of attention of numerous references in written Medieval sources, from the very moment of the Islamic conquest until the crisis of the Caliphate. Therefore, works oriented towards the reconstruction of the topography of Umayyad Córdoba has dedicated a good part of its attention to this building, reproducing the descriptions and data provided by successive authors. Using the information provided by the court scribes, it has been possible to establish a general framework of the palace complex, specifically its placement, the name of its gates, and some of its rooms, but it is quite unspecific when defining the details of the internal distribution and its architectural characteristics, which have been translated in unequal suggestions for the establishment of its limits and topographical reconstruction.

Rafael Castejón, Córdoba scholar and lover of Islamic archaeology, was the first to publish an extensive thesis based on the Caliphal Córdoba, which dedicates some paragraphs to the Umayyad Alcázar (Castejón y Martínez de Arizala 1929, pp. 279–80), based on the combination of the documentary information available and the direct knowledge of the material evidence recovered at various points of the capital. The result is a schematic plan

(Figure 2A) in which the southern limit of the Alcázar is located as coinciding with the line marked by the wall of the *qibla* from the expansion of Al-Hakam II in the Congregational Mosque. This façade would be defined by a strong wall documented in an excavation performed in 1922 by the Córdoba Archaeological Society in the Courtyard of Carruajes (or the Southern Courtyard) of the Episcopal Palace (Ibid., p. 279). It is, then, the first attempt of topographic reconstruction from the brief direct material information.

Outside the local realm, the contributions of Évariste Lévi-Provençal stand out, in a first chapter dedicated to *Cordoue, capitale du califat umaiyade d'Occident* ["Córdoba, capital of the western Umayyad caliphate"], in his book *l'Espagne musulmane au X$^e$ siècle* ["Muslim Spain in the 10th century"] (Lévi-Provençal 1932, pp. 221–24). This publication provides the first general traces of the palace complex, despite being aware of the restrictions and risks of these approaches, which were based exclusively on information from Arabic sources. Years later, the same author dedicated a section to "*Córdoba en el siglo X*" ["Córdoba in the 10th century"] in Volume V of *Historia de España* ["History of Spain"] edited by R. Menéndez Pidal (Figure 2B), in which the details related to the Umayyad palace are omitted, but he proposes a reconstruction of the *medina* in which he locates the Alcázar to the west of the mosque, separated from the southern wall of the city, a space where he locates the road or *al-Raṣīf* (Lévi-Provençal 1957, p. 235). In this monograph, Leopoldo Torres Balbás writes an extensive section on Umayyad art and architecture, in which he dedicates some lines to the Córdoba Alcázar, where he orderly copies the main information regarding the work undertaken by the successive Umayyad emirs and caliphs and lists the pavilions and spaces mentioned by the scribes (Torres Balbás 1957, pp. 590–94).

A work of particular value for understanding the Al-Andalus palace of Córdoba is the translation by Emilio García Gómez of *Anales Palatinos* by Ibn Hayyān, as this book details the events that occurred in Córdoba and, specifically, in the official residences of the caliph Al-Hakam II, "*the old Córdoba Alcázar or Madīnat al-Zahrā', or any of the royal orchards*" (García Gómez 1967, p. 31). This translation complements the prior study of the topography of Córdoba in the "Annals of Al-Hakam II", which systematises the information provided in these *Annals* regarding the Umayyad Alcázar in the Caliphate era (García Gómez 1965, pp. 322–34).

The lack of local archaeological studies on the Islamic era in Córdoba after the disappearance of these distinguished Arabists was covered by scholars and enthusiasts brought together at the Royal Academy of Córdoba. Accordingly, it is necessary to mention the contributions of Antonio Arjona, the main representative of this philological trend. Among his prolific works, he dedicated various texts to the Alcázar of Córdoba, in which he reproduces the translations of the main Arab texts (Arjona Castro 1997, 2001; Arjona Castro and Lope y Lope de Rego 2001, 2002) and proposes a reconstruction of the different areas that, according to this author, feature Córdoba palace (Arjona Castro and Lope y Lope de Rego 2001, p. 176) (Figure 2C).

In his prolific bibliography, Basilio Pavón Maldonado also addressed the question of the Andalusian Alcázar of Córdoba for those who proposed a reconstruction based on the texts provided by Islamic authors (Ibn Baškuwāl, mainly) and, specifically, in the hypothetical recreation of the southern façade of the city, the Bāb al-Sudda and the reef beside the gate to the bridge (Figure 2D) (Pavón Maldonado 1988, p. 170, Figure 1 and p. 186, Figures 9 and 10).

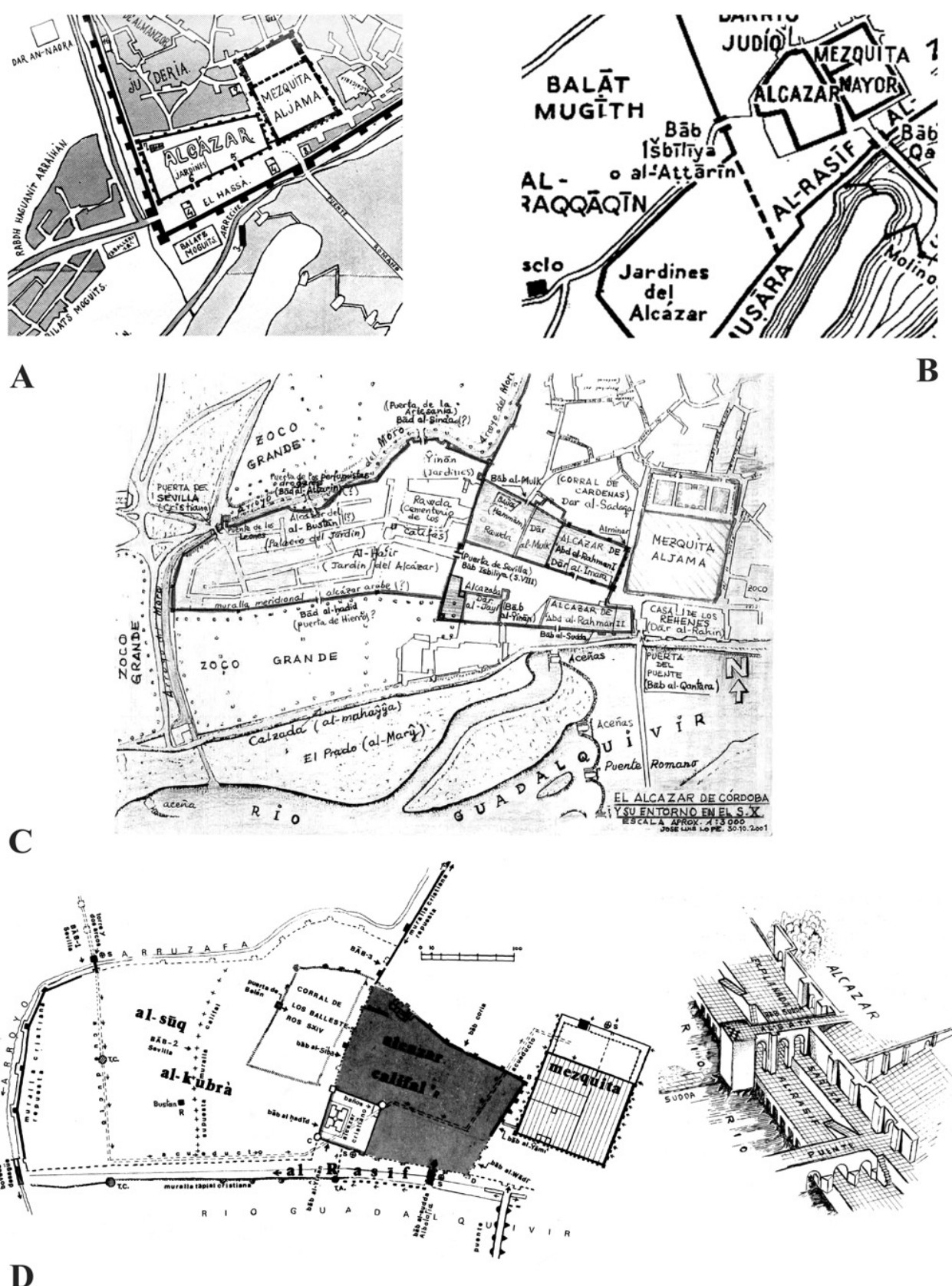

**Figure 2.** Hypothesis of the restoration of the Islamic Alcázar (Fortress) of Córdoba: (**A**) Castejón y Martínez de Arizala (1929); (**B**) Lévi-Provençal (1957); (**C**) Arjona Castro and Lope y Lope de Rego (2001); (**D**) Pavón Maldonado (1988).

Regarding the information provided by archaeology, until the end of the 20th century, excavations performed in the area have been scarce. The most extensive and well-known ones were for the recovery of the baths of Plaza Campo Santo de los Mártires between 1961 and 1964, conducted by Félix Hernández and Rafael Castejón (Castejón y Martínez de Arizala 1961–62a; Marfil Ruiz 2004, pp. 53–56; Murillo Redondo and León-Muñoz 2019, p. 134). On the same dates, small boreholes were opened in the Bishop's Gardens, within the old bishop's palace, with the aim of locating the graves of the caliphs (Castejón y Martínez de Arizala 1961–62b). A decade later, excavations were performed in the same gardens of the provincial library, in this case, aimed at the recovery and restoration of the north section of the wall of the Alcázar, by F. Hernández and with the co-ordination of Ana María Vicent, at that time, the director of the Archaeological Museum of Córdoba (Vicent Zaragoza 1973; Velasco García 2013).

The space that has had the most frequent intervention is the Alcázar de los Reyes Cristianos, fruit of the subsequent activities for the recovery of this high heritage value building. Thus, between 1951 and 1968, the municipal architect, Víctor Escribano Ucelay, carried out various archaeological explorations in the basement of the building including the western courtyard or "Women's Courtyard" so that a Congress building could be installed, and in the western courtyard (later known as "Mudéjar or Morisco Courtyard") to prepare it for tourist visits (Escribano Ucelay 1972, p. 74; León-Muñoz 2020a, p. 264). Soon afterwards, in 1974, Ana María Vicent and Alejandro Marcos conducted an excavation in the northern area of the Women's Courtyard, the results of which were unpublished when the work was interrupted due to a lack of financing. In 1981, J.F. Rodríguez Neila made a small intervention beside the south-east tower of the Alcázar for its reconstruction. Barely a decade later, in 1990, some test pits were opened to support its restoration, under F. Godoy and A. Ibáñez, as a prior phase for the preparation of a restoration project of the early-Medieval castle's wall sections. Unfortunately, the results of these activities were not published, and they hardly provided any information on the knowledge of the Al-Andalus palace complex (*cfr.* León-Muñoz 2020a, pp. 264–68).

The real inflection point in the research came at the end of the 20th century, with the studies undertaken by Alberto Montejo and José Antonio Garriguet, from an intervention of support for the restoration of Alcázar de los Reyes Cristianos (Garriguet and Montejo 1998). The research aim of the responsible parties of this activity more than surpassed the initial goals, restricted to the early-Medieval castle, and suggested an analysis of the Al-Andalus Alcázar complex and its urban surroundings based on the aforementioned written information and, above all, in the recovery and integration of the existing archaeological documentation up until that moment[1] (Montejo Córdoba and Garriguet Mata 1998; Montejo Córdoba et al. 1998, 1999). The combination of all of this information allows for the suggestion of a new hypothesis of for the restitution of the palace complex, whose western and southern limits coincided with those of the urban wall, a proposal for the location of its gates, and the internal distribution of the main functional areas (funeral, residential and military) (Figure 3A) (Montejo Córdoba and Garriguet Mata 1998, p. 326, plano 2; Montejo Córdoba et al. 1998, p. 9). Currently, this hypothesis has become the most recognised and rigorous one, to which light modifications have been included, based on new archaeological discoveries.

Since the beginning of the 21st century, within the framework of the Agreement between the Municipal Management of Urban Planning and the Archaeology Area of the University of Córdoba (GMU-UCO Agreement), various activities in the space occupied by the Alcázar and its immediate surroundings have taken place. The first of these, led by Silvia Carmona, consisted of the opening of various probes into the wall of the Orchard of the Alcázar and allowed for the documentation of the emir's promenade, which held the paved forecourt (or *al-Raṣīf*) running to the south of the Alcázar (Murillo Redondo et al. 2009–10; Carmona Berenguer 2020).

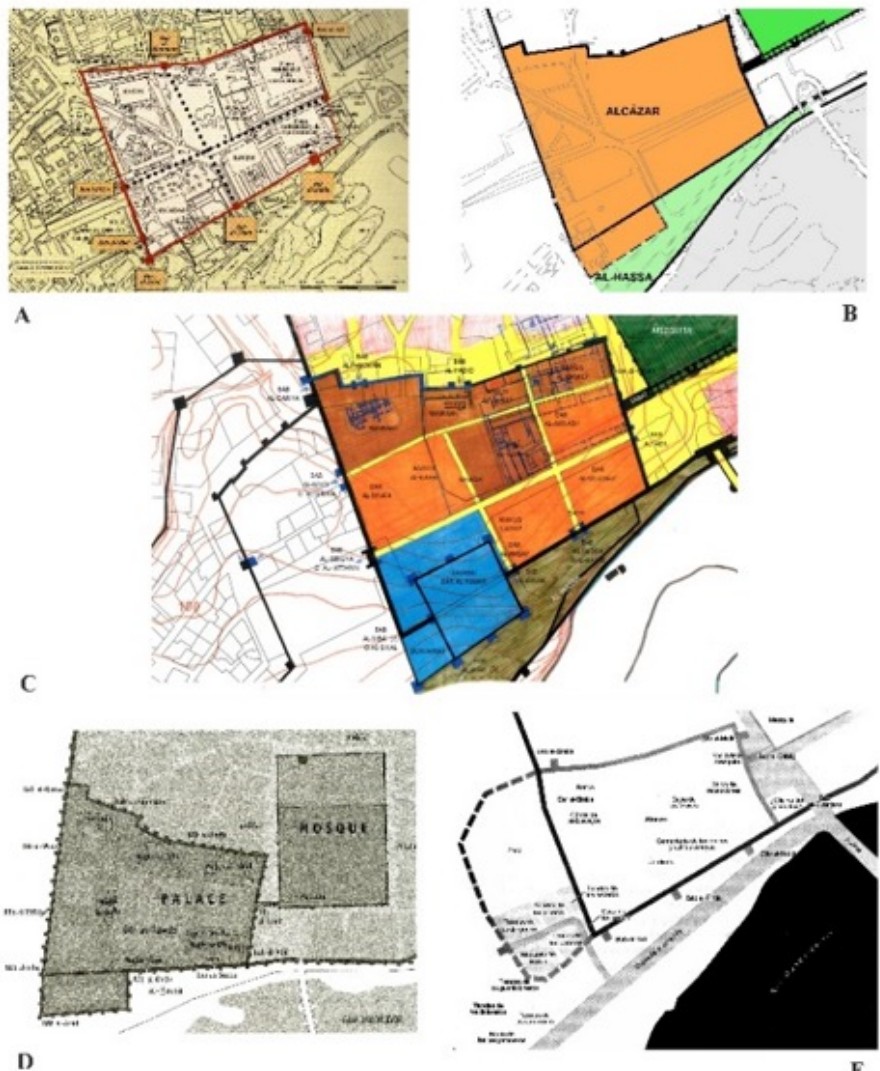

**Figure 3.** Hypothesis of the restoration of the Islamic Alcázar (Fortress) of Córdoba: (**A**) Montejo Córdoba and Garriguet Mata (1998); (**B**) León-Muñoz and Murillo Redondo (2009); (**C**) Murillo Redondo and León-Muñoz (2019); (**D**) Arnold (2017); (**E**) Manzano Moreno (2019).

The following archaeological intervention performed between 2002 and 2004 focused once again on the interior of the Alcázar de los Reyes Cristianos, specifically, in the so-called "Women's Courtyard". The new information obtained from this excavation allowed them to find more of the origin of the Al-Andalus Alcázar and reconstruct a complete sequence of occupation, with special interest in the Late Antiquity, Umayyad, and Almohavid (León-Muñoz and Murillo Redondo 2009; León-Muñoz et al. 2008; León-Muñoz 2013; León-Muñoz 2020a, among others). As for the main information regarding the limits of the complex, the identification of the eastern closing wall of a late-antique enclosure advanced on the line of the southern wall of the city stands out, which has been preserved as a guiding element for later constructions until the end of the Middle Ages (Figure 3B). On the other hand, the results allowed the authors to specify the great transformations the building experienced during the last quarter of the 12th century, when an Almohad palace was erected and its area expanded for its transformation into an extensive *alcazaba* (León-Muñoz and Murillo Redondo 2009; León-Muñoz 2013; Murillo Redondo 2020). The verification of the stratigraphic complexity of the architectural complex formed by the different areas that define the Alcázar has shown evidence for the need to address successive studies through a significant change in perspective; that is to say, in addition to topographic

approximations, it would have to focus attention on the diachronic reading, as it is a space of prolonged occupation and continuous transformation. We believe that this is one of our main contributions to the investigation of the Islamic Alcázar of Córdoba.

Regarding the northern limit of the Umayyad Alcázar, the archaeological intervention performed in the "Garaje Alcázar" (currently "Parking La Mezquita") at C/Cairuán, no. 1, served to document part of the north wall with a specific zig-zag pattern, associated with an internal access door to the palace area (Vargas Cantos et al. 2010). To the south of this wall, in 1993 and afterwards between 2000 and 2002, two archaeological interventions were carried out into the Caliphal Baths, as support for converting it into a museum and highlighting the importance of this building, which was initially discovered at the beginning of the 20th century and was the subject of a later excavation at the beginning of the 1960s (Marfil Ruiz 2004, p. 53). Recent activities have confirmed the data provided by the excavation in the 60s and the later studies of the decoration on plaster fragments (*cfr.* Ocaña Jiménez 1984, 1990); that is, the floor of the caliphal bath, with subsequent stages of renovations and expansion such as the "ante room" of the Taifa era and the Almohad bath located to the west of the previous ones (Marfil Ruiz 2004, p. 59).

Finally, the conditioning work for some of the premises of the Episcopal Palace has favoured the completion of archaeological activities of interest such as the parament reading in the western façade bordering the Congregational Mosque (Velasco García et al. 2012) and the excavations in 2008 in the southern courtyard (also known as the "Patio de Carruajes") (Marfil Ruiz 2010, pp. 463–64; Ruiz Diz 2010). In their work, Marfil preserves, generally, the reconstructive hypothesis proposed by R. Castejón, by coinciding the limit of the Alcázar with "*the tower of the Southeast corner of the episcopal palace, outlining the southern façade of this palace in relation to its baroque courtyard. The pavement or al-Hassa was between the southern wall of the Caliphate Alcázar and the southern wall of the medina, so we do not agree with the hypothesis that defends the Alcázar reaching the wall of the medina in its southern limit*" (Marfil Ruiz 2004, p. 57). This interpretation has been definitively dismantled with the recent results provided by the last intervention in the same courtyard between 2015 and 2019 (Ortiz Urbano 2022).

The quality and relevance of the archaeological information recovered in the last two decades have once again brought attention to the palace complex, which held the centre of power in Al-Andalus, making it a protagonist for research in its most diverse aspects. The integration of these data has allowed for the suggestion of a new reconstructive desing for the plan of the Alcázar (Murillo Redondo and León-Muñoz 2019; Murillo Redondo 2020), and specifically, the internal organisation of the complex (Figure 3C). This hypothesis has been the inspiration for the preparation of new planimetrics that illustrate one of the few international studies on Islamic palace architecture in the western Mediterranean in which the Umayyad Alcázar of Córdoba is already mentioned (Arnold 2017, p. 21, Figure 1.6) (Figure 3D)[2].

The most recent planimetric proposal is based, once again, on the information provided by written sources, specifically the *Anales Palatinos* by Ibn Hayyān (Manzano Moreno 2019) (Figure 3E). This hypothesis focuses on the question of the western closure of the Alcázar and the extension of the urban wall, and specifically, in the location of the Gate of Seville. For most authors, this door was opened in the western section of the palace complex, whose layout would coincide with that of the wall of the Medina. According to Manzano, "*the sources, however, make it clear that there was a space between the Alcázar and this gate of Seville, occupied by the market*" (Ibid., p. 419, nota 35). In order to do this, it refers to the passage in the *Anales Palatinos* that tells of the transfer of the caliph Al-Hakam II from the Alcázar of Madīnat al-Zahrā' to the palace of Córdoba in March 975[3]. However, as García Gómez recognises, in this text, "*we are not told, however, which door he used to enter the city*" (García Gómez 1965, p. 349), nor does it refer to the aforementioned gate of Seville or Bāb al-'Aṭṭārīn, the point where, according to García Gómez and E. Manzano, the retinue would enter the market and from there the Alcázar, by the Gate of Iron (or New Gate), in the southern façade. He proposes, then, the possibility of moving the gate of Seville

further west, into a hypothetical larger walled area. To a certain extent, this coincides with the hypothesis of B. Pavón, by locating a possible caliph wall between the *Alcázar* and the early-Medieval expansion of the Córdoba walled area (Pavón Maldonado 1988, *vid. supra*).

The growing interest in this palace complex has been expressed through congresses organised by the Royal Academy of Córdoba in 2020, in which the need to address systematic research projects on the Alcázar was highlighted and the main archaeological novelties were presented, published in a monograph issue of the magazine Al-Mulk published by the Institute of Caliph Studies (cfr. León-Muñoz 2020a, 2020c, among others).

Definitively, as we see, the majority of attempts to reconstruct the Al-Andalus Alcázar have mainly been based on a limited number of textual references, focused on the Umayyad era. From the middle of the 11th century, the palace complex disappears from Islamic chronicles, losing the protagonism that it once had as a centre from which the politics of Al-Andalus was ruled. The later evolution of the Al-Andalus Alcázar has been practically overlooked in the historiography until the moment of the Castilian conquest in 1236. These written sources transmit a static image of the complex, that is to say, a building whose limits were established from the moment of the conquest, and that remained stable during the entire era of Islamic rule in Córdoba. Thus, it has been assumed that the Alcázar that Fernando III appropriated should be similar to that occupied by 'Abd al-Raḥmān I in the second half of the 8th century, with the exception of the regular contributions made by successive Umayyad emirs and caliphs. However, the archaeological evidence shows a richer and more complex panorama, in which subsequent architectural transformations occurred, characteristic of the space of high symbolic and propaganda value, as a seat of civil power throughout the Middle Ages. For all these reasons, it is essential to carry out a diachronic journey of the transformations experienced by this architectural complex.

### 3. The Precedents: The Late Antiquity Civil Complex

The location of the Umayyad Alcázar in the south-western sector of Córdoba is not a coincidence. The seat of Al-Andalus authority was established on the buildings that represented Visigoth civil power, in a common phenomenon of the symbolic and functional appropriation of these spaces by the subsequent rulers of the city. The written sources mention the palace that the Islamic governors (walis) seized after the conquest of the city. The (*Ajbār Maŷmūʿa* 1867) (ed. Lafuente and Alcántara) refers to this building as "the palace of Córdoba" (Ibid., p. 12), where Mugīt al-Rūmī "lodged" upon his arrival in the city (Ibid., pp. 14–27), and from which he was dethroned by Mūsà ibn Nuṣayr as the new governor. Al-Maqqari, following Ibn Baškuwāl, calls it *Balāṭ Ludrīq*, but "*not because he built or founded it, given that it was the product of kings who had preceded him, and the place where they stayed when they came to Córdoba, but because the Arabs, ignoring the name of the founder, after defeating Rodrigo, called it as such, due to aforementioned monarch having stayed there*" (*Nafḥ al-ṭīb* by al-Maqqari; taken from Arjona Castro and Lope y Lope de Rego 2001, pp. 154–55). Beyond the legendary origin attributed to the palace, these references intend to expressly show the age of the building, prior to occupation by the Visigoth monarchs ("*primitively inhabited by the infidel kings who ruled the country since the time of Moses*"), something which, on the other hand, granted a certain prestige: "*whose interior, as well as the buildings that surrounded it, were full of primitive constructions of Greeks, Romans and Goths and other extinct peoples*" (Al-Maqqarī 1855–60, pp. 302–3; translation by Arjona Castro 1982, p. 207, doc. 272).

As the archaeological information shows, the transfer of power spaces to this south-western sector from its location in the northern sector of the Roman *urbs* and the forming of the "Late Antiquity civil complex" began during the 4th century, at a time yet to be defined more precisely, and was clearly established in the 5th century. The motive of this change is fundamentally explained by the advantageous conditions offered by this strategic placement, which allowed for direct control of the river, the bridge, and the entire sector linked to the port and commercial activity (*cfr.* León-Muñoz and Murillo Redondo 2009).

The historiography tradition has assumed that the "Visigoth palace" was an isolated building inside the walled area, in the northern sector of the ground later occupied by the

Islamic Alcázar, alongside the placement of the Congregational Mosque and separated from it by just one street, interpreted as the fossilisation of the *cardo maximus* of the Roman city (Marfil Ruiz 2000, pp. 129–30). This proposal is based on a vague textual reference in the *Ajbār Maŷmū´a*, according to which, during the confrontation between 'Abd al-Raḥmān I and Yūsuf al-Fihrī, "*Abú Otsmen was besieged in the tower of the main mosque, which was in the Alcázar, and obligated to surrender, on the condition that he would not fight; however, he placed shackles and took him prisoner*" (*Ajbār Maŷmū´a* 1867, p. 88). Despite this evidence, M. Ocaña deduces that "*a building near to the palace had been provisionally enabled for congregational mosque, and that one of the towers of the palace* [perhaps the eastern section], *due to its dominant position, served as the sawmu'a of the mosque*" (Ocaña Jiménez 1942, p. 350). By extension, it has been deduced that the old Visigoth palace would have been in the surroundings of this first oratory. This hypothesis has a weak archaeological basis; specifically, after the localisation of the Late Antiquity episcopal basilica under the oratory of 'Abd al-Raḥmān I (León-Muñoz and Ortiz Urbano, forthcoming). Be that as it may, the intramural location of this palace, which is at a considerable distance from the line of the southern wall and the bridge, as this hypothesis holds, would be counter-productive, as it would cancel or, at least, considerably restrict the possibilities of the direct control of the vital elements of the city throughout history and that determined one of the most substantial transformations of the urban image of Córdoba during the Late Antiquity.

On the other hand, this concept of the palace as a single building is becoming more nuanced with more recent suggestions in the research and archaeological evidence recovered in recent decades in this south-western section of the city. Regarding the concept of a palace, as has been proposed for cities such as Toledo or Reccopolis, it is not only understood as a residence, *"but as a defining complex of the power structure perfectly represented in its political-ideological aspects"* (Olmo Enciso 1987, p. 352), which would imply the existence of a residence, an administrative seat, guard stations, court areas, one or more basilicas, etc.[4]. Similarly, the visual description provided by Sidonio Apolinar of the dependencies of the palatial area of Toulouse under Ostrogoth control, in the middle of the 5th century CE, references the existence of various buildings, among them a great palace with courtrooms and reception halls, a building for treasure, residential constructions, stables, and a chapel (Guyon 2000, p. 236).

The most recent archaeological data point to the existence of a complex of structures of notable architectural entity and of different functionality—representation, residential, military and administrative—distributed throughout this urban sector[5], which we have come to call the "civil complex" (*cfr.* León-Muñoz and Murillo Redondo 2009, 2014), in the same way as the constructions linked to the bishop would be in an episcopal complex. Some of them, perhaps the *"aula regia"* type, might have been the famous "Rodrigo's palace", whose identification is currently a simple chimera.

Structures belonging to this "civil complex" have been located at two points.

In the South Courtyard of the Episcopal Palace, the interventions carried out between 2015 and 2019 by Enrique León and Raimundo Ortiz have provided an interesting sequence of ground occupancy. The existence of an important building from Late Antiquity stands out, whose eastern closure is defined by a great wall of more than two and a half metres in width, made with calcarenite masonry and architectural elements that have been re-used from the Roman age, whose external parament was later covered by the walls that define the eastern façade of the Umayyad Alcázar built in the era of 'Abd al-Raḥmān II (Figure 4). This wall would mark the western limit of the street, while the western closure would coincide with the gantry of the building at the end of the 6th century documented in the Courtyard of the Oranges in the Mezquita, with a parallel layout (León-Muñoz and Ortiz Urbano 2023, p. 171). The adaptation of the Al-Andalus Alcázar for this pre-existing building explains its divergent direction from the western façade of the Congregational Mosque at this point. The southern closure of this building could be defined by a thick wall perpendicular to the previous one, parallel to the river in a NE–SW direction, which only a part of a bay is preserved. This wall is contained inside a

structure of early Emirate chronology, cut in the modern era for the installation of sanitation infrastructure in the courtyard. In the 2008 excavation, this structure was identified by its type and the bond of the stone blocks as the remains of the wall of the pre-Islamic era with its southern face covered by an emir wall, attributed to ʿAbd al-Raḥmān I, which, according to this hypothesis, would mark the southern limit of the Umayyad palace (Marfil Ruiz 2010, pp. 463–64). As we shall see, the evidence recovered in other areas of the courtyard has allowed us to dismiss this last proposal.

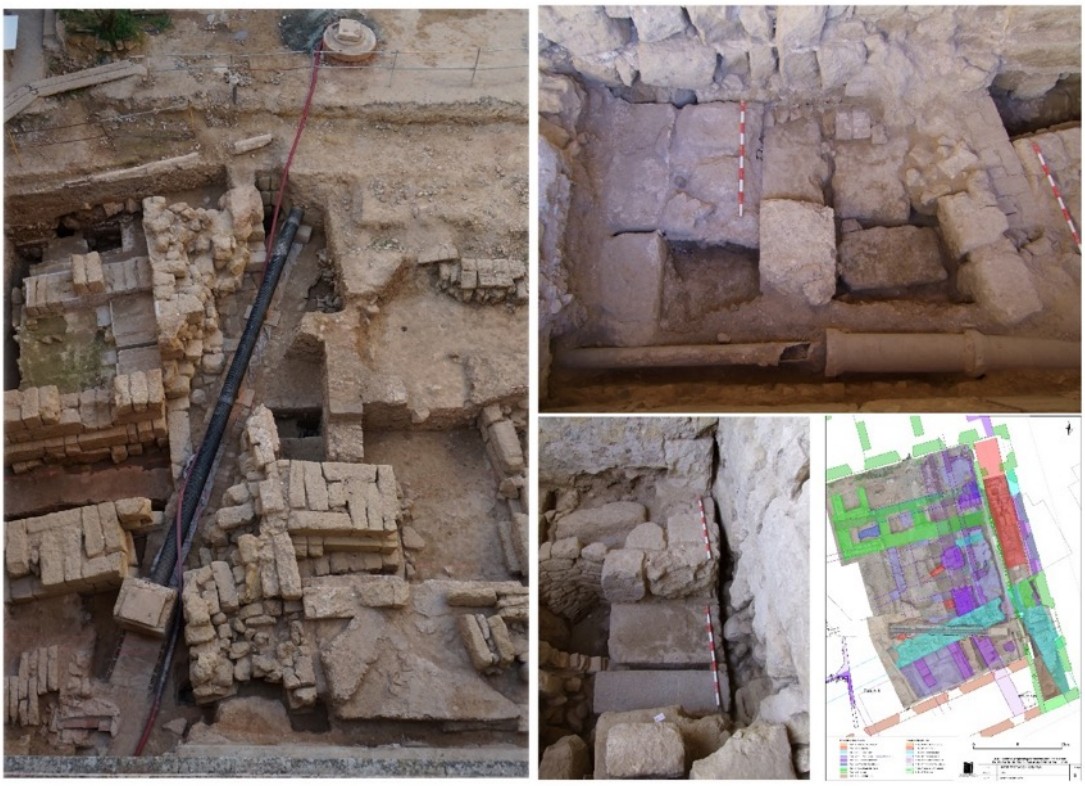

**Figure 4.** Late Antiquity structures documented on the Patio Sur del Palacio Episcopal (South Courtyard of the Episcopal Palace) (image: Raimundo Ortiz Urbano). In red, the structures belonging to the Late Antique Period.

In the Women's Courtyard of the Alcázar de los Reyes Cristianos, the results of the archaeological intervention led by A. León between 2002 and 2004 were especially revealing in terms of the existence of this Late Antiquity "civil complex"; more specifically, with the so-called *Castellum* (cfr. León-Muñoz and Murillo Redondo 2009, 2014). Advanced regarding the line of the southern wall of the Roman city, a walled area was put up with masonry, dated to the 5th century CE, which preserves part of the eastern section, being preserved for centuries as the south-east closure of the Alcázar. This area would have extended to the south of the early Medieval castle, pursuant to the evidence provided by a 1981 intervention (León-Muñoz 2020a, p. 283)[6]. As a working hypothesis, a rectangular plant has been proposed up to the south-east section of the urban wall, whose confirmation depends on subsequent archaeological activities in the current Avda. del Alcázar.

In the interior of this area, structures of a notable monumental nature are set:

On the one hand, two rows of parallel columns are documented (Figure 5A), in a NE–SW direction, all of them reused, which were equally included in the previous Umayyad walls. In addition to these two excavated rows in the interior of the Women's Courtyard, a third row has been identified, perpendicular to the previous one, reused in the alignment of the western closure wall of the Christian Álcazar; which, at the same time, increases the walls of an Almohad palace (León-Muñoz and Murillo Redondo 2009, p. 411, Figure 3).

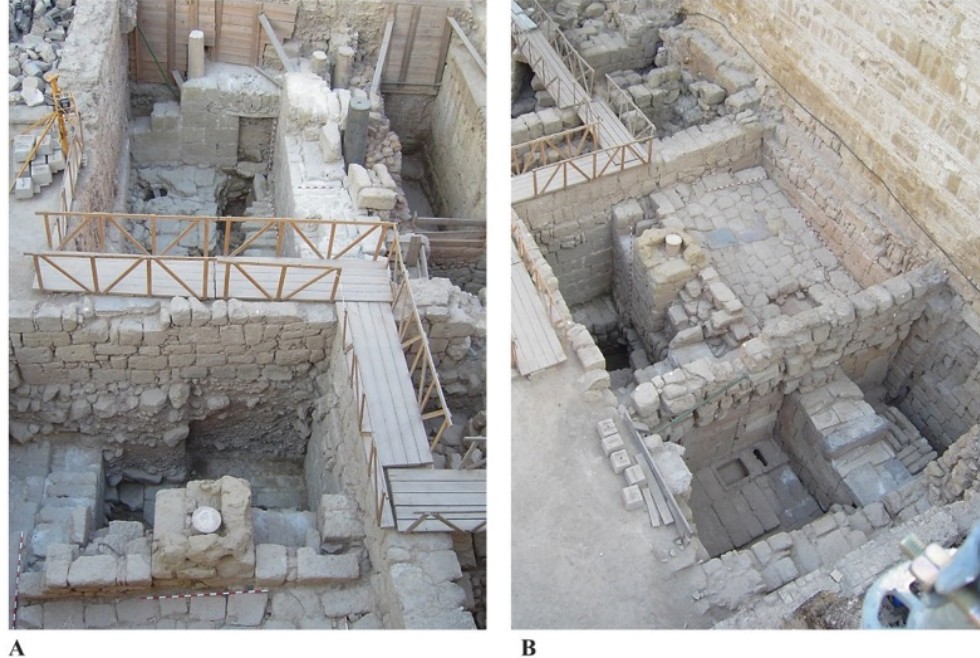

**Figure 5.** Late Antiquity structures documented on the Patio de Mujeres del Alcázar de los Reyes Cristianos (Women's Courtyard of the Christian Monarcs' Fortress): (**A**) two rows of parallel columns reused NE–SW direction; (**B**) walls of compartmentalisation of spaces, where various gates would have been opened to access the different rooms set in its interior (image: GMU-UCO Agreement).

A second element documented is the floor level associated with the columns, formed by a slab of limestone in which the print of a column shaft is marked and a slab of lime mortar.

On the other hand, although we cannot stratigraphically associate them as they do not have any direct physical connection between them, we could document some powerful walls of the compartmentalisation of spaces, performed with re-used masonry, but carved well, which preserves the axis marked by the row of columns where various gates would have opened to access the different rooms set in its interior (Figure 5B). We believe that these structures correspond to a more advanced period, since they are endorsed to the interior of the previously mentioned western section that defines the area of the 5th century, in whose interior an intense remodelling has been performed. As such, we have a clear relative sequence, although one with wide chronological limits.

According to this evidence, we believe that this area must have been occupied by a complex of buildings of architectural entity with different functionalities, but linked to the civil authorities of the city, where, at the same time, the new Muslim rulers were installed, as shown by the occupation sequences documented at various points of this urban sector. This fact entails an important novelty when proposing the origin and evolution of the Al-Andalus Alcázar. From the archaeological information available, its origin can be seen to be more complicated than previously imagined, being able to locate the germs of the palatial architectural complex in this *castellum* or extramural area with strategic placement.

## 4. The Evolution of the Al-Andalus Palatial Complex

### 4.1. The Period of the Emirate

Above this this complex of buildings, the subsequent governors of the Umayyad dynasty carried out architectural interventions that totally changed its original physiognomy, while being heavily conditioned by the previous constructions. The Alcázar is a palatial and military complex formed over five hundred years of Islamic presence in Córdoba. The subsequent renovations in the Umayyad era and, in particular, the intense architectural remodelling of the Almohad era have transformed the building, though in a way that

draws attention to it, always preserving and re-growing the elevations of a part of the Late Antiquity building.

Therefore, one of the pending tasks in the research is to try to specify the architectural evolution of the Alcázar from the combination of written information and material evidence. However, as we have already indicated, the documentary sources do not describe the structures in enough detail to identify them with the excavated archaeological vestiges.

Traditionally, the repair of the walls of Córdoba in 766 (Arjona Castro and Lope y Lope de Rego 2001, p. 155; Acién and Vallejo 1998, p. 113) and the later construction of the Alcázar in 784 has been attributed to Abd al-Raḥmān I (Lévi-Provençal 1932, p. 222; Ocaña Jiménez 1935, p. 165; Torres Balbás 1957, p. 591, among others), coinciding with the installation of the first state institutions such as the House of the Post Office (Acién and Vallejo 1998, p. 115; García Gómez 1967, p. 87). However, it seems that the first great fortification of the Alcázar was due to the climate of social agitation in the capital during the emirate of Al-Ḥakam I. Specifically, in the year 805 (198 H.): *"Waiting then for his rebellion, he prepared for the uprising, directing the repair and reinforcement of the walls of Córdoba, reconstructing the damaged parts, which he performed with much envy and vigilance, until doing it as he wanted, achieving that it were totally impregnable; he later directed the part of the pit that was missing to be dug around it, strengthening it"* (Ibn Ḥayyān 2001, al-Muqtabis, II, 1 [98v], p. 43 of the trans.). According to al-Maqqari, in this same process, *"he took precautions, repairing the walls of the city of Córdoba and closing its breaches, paving the path to its Alcázar (. . .) providing ammunition and supplies, fortifying the Alcázar on all sides, repairing its doors and closures, reinforcing weak points and multiplying chamberlains and guards"* (Al-Maqqarī 1855–60, p. 380, taken from Arjona Castro and Lope y Lope de Rego 2001, p. 158).

Regarding the archaeological evidence, it is necessary to consider the difficulties when determining its chronology in a specific moment of the 8th century. This is the case of the section base of the wall documented in the ground of the current "Parking la Mezquita", whose internal parament was constructed with a technique of *opus africanum*, consisting of vertical linked blocks of stone and masonry filling with abundant lime mortar. The use of this building technique of evident Late Antiquity antecedents has allowed the association of this structure with the reconstruction of the walled area of the *medina* undertaken by Abd al-Raḥmān I in 766 (León-Muñoz and Montejo Córdoba 2023, pp. 192–93) or perhaps with the generalised fortification made by Al-Hakam I.

As regards to the interior of the Alcázar, from the evidence recovered in the Women's Courtyard, the first interventions documented consist of the reuse and repair of the Late Antiquity structures with the eventual addition of some internal construction. Specifically, the regularisation of the old line of urban wall (Figure 6), made with reused materials, and the compartmentalisation of some interior spaces with masonry walls raised directly above the Late Antiquity levels of the 7th century. In the South Courtyard of the Episcopal Palace, as we have already indicated, in the initial phase of the Alcázar, the southern border wall of the Late Antiquity building was wrapped with transporting materials or with the alternation between masonry and stone blocks, which presents analogies with the walls that restrict the *mida'a* of Hisham I to the east of the Congregational Mosque.

The great renovations of the entire complex were performed in the emirate of 'Abd al-Raḥmān II in the second quarter of the 9th century, as shown by the notifications provided by written sources and archaeological evidence. This ambitious architectural program is a direct reflection in the capital of the new political model that this emir was trying to impose in Al-Andalus and is stated as such by Ibn Ḥayyān in various paragraphs of his chronicle:

> "According to Ahmad b. Muhammad Arrazi: The Emir Abdarrahman b. Al-hakam was the first of the Marwan caliphs to give prestige to the monarchy of Alandalús, he gave it the pomp of majesty and conferred it a reverential nature, (. . .), he built Alcázars, he performed works, built bridges, brought fresh water to the Alcázar from the mountain tops (. . .). He made the terrace that dominates the main gate of the caliph's Alcázar, the first southern one, called Gate of the Azuda (Bab assuddah), placing it on top like a crown, which sealed

its extraordinary grandeur (. . .) He was also the one who built the promenade to the banks of the river in the south-western part of the Alcázar, extending it from the eastern section of the city, to the end of the western section of the Alcázar, adding an extension to this section which connects it to the edge of the great market of Córdoba, and leaving the hill called Abu 'Abdah at the gate of Arsenal (=Bab assina'ah), the northern one between the Alcázar, within which he also made great buildings and marvelous works that are attributed to him" (Ibn Ḥayyān 2001, al-Muqtabis, II,1 [140r], pp. 171–72 from trans.).

"According to Isa b. Ahmad Arrazi: It was the emir Abdarrahman who built the treasury to the gates of the Alcázar, through the external part, and placed a team of four treasurers inside it (. . .). He was the first to build lavish buildings and generous Alcázars, using advanced machinery and traveling through all the regions in the search for columns, finding all the instruments of Alandalús and bringing them to the caliph's residence in Córdoba, so that every famous construction there was his own building and design" (Ibn Ḥayyān 2001, al-Muqtabis, II,1 [143v], p. 181 from trans.).

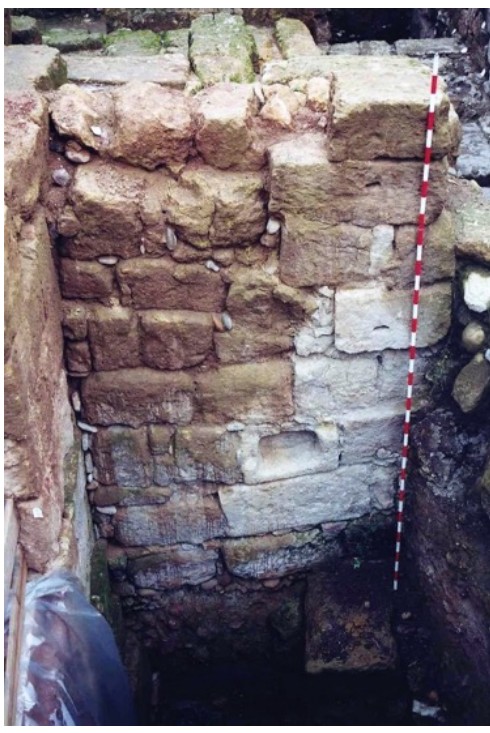

**Figure 6.** Structures of the early Emiral period (8th century) in the Patio de Mujeres del Alcázar de los Reyes Cristianos (Women's Courtyard of the Christian Monarcs' Fortress), consisting of the regularisation of the former urban wall line made with reused materials (image: GMU-UCO Agreement).

Among the interior buildings attributed to him, two are mentioned: the "Ministerial house" (*Bayt alwizārah*) (Ibn Ḥayyān 2001, al-Muqtabis, II,1 [144r], pp. 183–84 from trans.) and the "Pavilion of Joy" (=*Dār assurūr*) (Ibid. [164v], p. 247 from trans.)

In this case, the archaeological evidence appears to coincide with the information provided by the great Al-Andalus chronicler. The different archaeological activities conducted at various points of the walled area of the Alcázar have allowed us to confirm, from the arguments of a different nature and force, that during the first half of the 9th century, the perimeter of the palatial complex was definitively defined or, at least, in its eastern, northern, and the surroundings of its southern front, which would be preserved, without great modifications, until the end of the Caliphate.

On the one hand, a reading of the elevations of the eastern façade of the episcopal palace recorded by P. Marfil has served to identify various sections with Umayyad paraments of stretcher and header blocks, similar to the bonds preserved in the first expansion of the Congregational Mosque located in front of the palatial complex. From the stratigraphic reading and the constructive characteristics, this eastern façade of the Alcázar has been dated to the era of the aforementioned emir (Velasco García et al. 2012, pp. 1912–13; Marfil Ruiz 2010, p. 461). A similar argument has been used to propose a coeval chronology for some of the flanking towers of the northern front of the Alcázar, which display a bossed masonry bond. This is a typical construction of some works during the emirate of ʻAbd al-Raḥmān II such as the external parament of the *mihrab* of the Córdoba Congregational Mosque (*cfr.* León-Muñoz 2020b, pp. 170–72; León-Muñoz and Montejo Córdoba 2023, p. 204).

In this northern section, during the archaeological intervention carried out in 1971 under the guidance of Ana Mª Vicent and the supervision of Félix Hernández in the Bishop's Gardens (Vicent Zaragoza 1973, p. 25), a stretch of the northern wall of the Alcázar was discovered, where a small door opens between two towers that lead to a parapet or "service walk" paved with irregular flat slabs, marked by a wall with a parallel layout of small towers or buttresses in its northern parament. To the east of this door, a perpendicular wall is set in which, at the same time, two bays of a monumental nature are open (Figure 7). All of these structures show ashlar masonry similar to those identified in other points of the complex and they enable the hypothesis of a chronology of the emirate period for its construction, while awaiting a more detailed and exhaustive study.

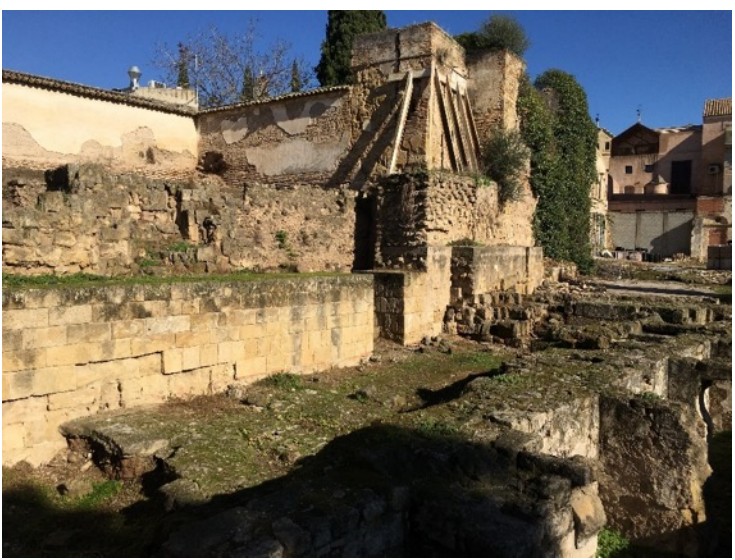

**Figure 7.** Northern enclosing wall of the Andalusian fortress documented on the Jardines del Obispo (Bishop's Gardens) (Image Alberto León).

In the South Courtyard of the Episcopal Palace, the intervention led by Raimundo Ortiz enabled the identification of the internal parament of the eastern façade of the Alcázar, which consists of the lining of the Late Antiquity building with a regular ashlar masonry wall of great size. Similarly, the area expands to the south with a strong structure made with header blocks, which sealed off the street that led south in an east–west direction, whose layout would coincide with the location of one of the eastern gates of the Al-Andalus Alcázar, for which its identification with the *Bāb al-Adl*, or the Gate of Justice (Ortiz Urbano 2022, p. 181) has been proposed. These details confirm the extension of the emir's Alcázar to the south line of the wall of the Medina (cfr. Montejo Córdoba and Garriguet Mata 1998) and requires discarding the proposals that closed the Umayyad palace around the current street called Amador de los Ríos (cfr. Arjona Castro and Lope y Lope de Rego 2001; Marfil Ruiz 2010).

On the other hand, the archaeological excavations performed at several points of the southern area of the Alcázar have provided a date based on the stratigraphic, type, material, and documentary arguments.

In the Women's Courtyard, renovations were identified in the Late Antiquity building, dated to the emirate of ʿAbd al-Raḥmān II, which suggests the remodelling of the interior of the *castellum*. On the one hand, it closes the spaces between the columns and shields the open bays in the Late Antiquity walls. To the south of successive lines of Roman walls, a solid ashlar masonry wall is raised, whose internal face, made from masonry, aimed to contain a strong clay filling[7]. Its external parament is supplied with stone block buttresses designed to contain the fillings and pressure of the sloping land. To the south of this façade, the space previously open has been reorganised, with a courtyard containing a well and a little water tank, with a square floor room to the west from which two gates that nearly face each other in the eastern and western sides have been documented, respectively (León-Muñoz 2022, p. 293) (Figure 8).

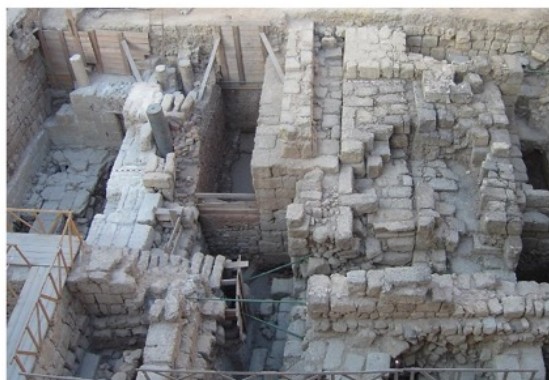
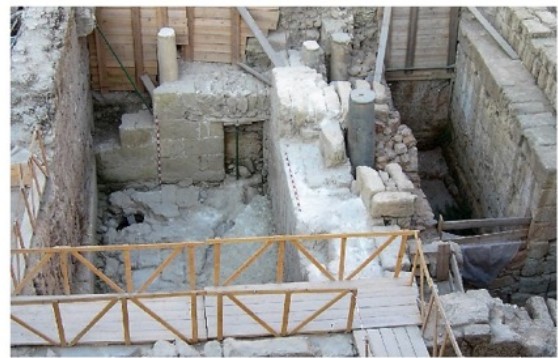
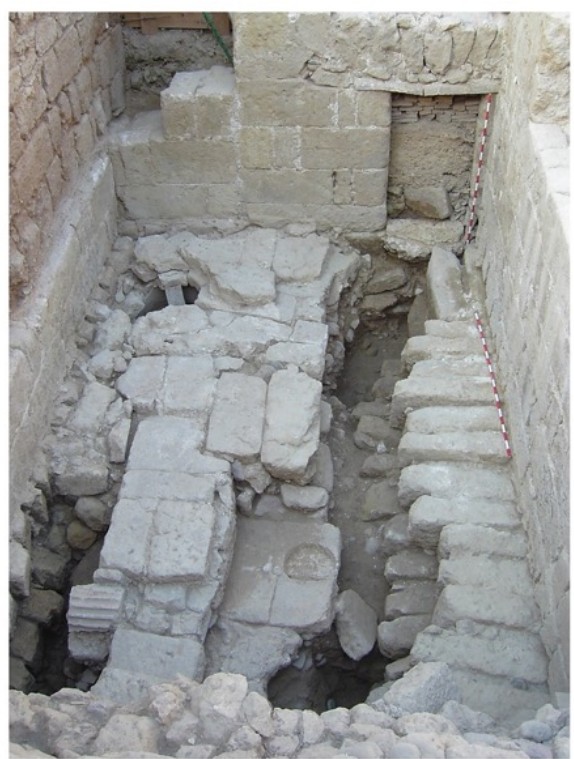

**Figure 8.** Structures dated during the emirate of ʿAbd al-Raḥmān II, documented on the Patio de Mujeres del Alcázar de los Reyes Cristianos (Women's Courtyard of the Christian Monarcs' Fortress) (image: GMU-UCO Agreement).

This ambitious building program not only affected the limits and internal distribution of the area, but it also implied the monumentalisation of the immediate surroundings, specifically, its southern façade, where the *Bāb al-Sudda*, the main door of the Alcázar, is located, also remodelled by ʿAbd al-Raḥmān II (Ibn Ḥayyān 2001, Muqtabis II-1, p. 172). This was confirmed thanks to the preventive archaeology activity led by Silvia Carmona at the wall of the Orchard of the Alcázar. This excavation enabled the documentation of a strong structure made with headed brickwork in its external façade and an internal parament made with an imitation of the *opus africanum* technique, in which specific boxes alternate with pillars of calcarenite brickwork and a masonry filling is joined with lime mortar. The location of this structure, the constructive characteristics, and the materials recovered from the associated fillings have enabled the identification of this with the promenade mentioned

by Ibn Ḥayyān, raised in the year 827–828, on which the paved road or *al-Raṣīf* was installed (Murillo Redondo et al. 2009–10, pp. 186–89; Carmona Berenguer 2020, p. 336).

The result of this ambitious architectural project is an extensive walled area that integrates and renovates the old buildings of the Late Antiquity era and the first stage of the independent emirate. Accordingly, M. Ocaña's concept is quite suggestive regarding the evolution of the Alcázar: *"it consisted of different quṣūr or palaces, built, within the walled area of the old mansion of the emirs, in different eras and by different princes. At the beginning, they would be isolated buildings but, little by little, by increasing the buildings and linking them between each other, they became mere quarters of the great Qaṣr Kabir, considered as a unit"* (Ocaña Jiménez 1935, p. 165). According to the author, the details indicate that it was during the government of ʿAbd al-Raḥmān II that the palatial complex was defined as a single architectural complex.

As is the case of the Congregational Mosque, successive emirs continued with works in the complex such as those mentioned in the Bayān al-Mugrib, referring to the "numerous buildings in the great Alcázar" that Muhammad I completed in the year 864–865 (Souto Lasala 1995, p. 221). Proof of the existence of the eastern closure of the area from the middle of the 9th century is the construction during the emirate of ʿAbd Allāh of the *sābāṭ* (Pizarro Berengena 2013, p. 234), which connected the Alcázar with the *maqsurah* of the mosque of ʿAbd al-Raḥmān II: *"it was he who built the sābāṭ between the Alcázar and the mosque of Córdoba to perform the prayers as a community. He ruled on the injustices of the administration and received complaints while seated before the door of the Alcázar* [Bāb al-Sābāṭ or Passage Door]; *the most powerful and the most humble could visit him"* (*Ḏikr balād al-Andalus* 1983, [91], p. 163 from the trans.).

*4.2. The Period of the Caliphate*

The construction of Madīnat al-Zaḥrā monopolised the attention of the Al-Andalus chronicles during the caliphate, so references to the Alcázar of Córdoba in these sources are even more scarce. However, as the former seat of Umayyad power in the capital, the caliphs preserved building activity, particularly during the government of ʿAbd al-Raḥmān III: *"One of his merits is that he did not leave in the Alcázar, between the works of his grandparents and the oldest surviving pieces, a single building to be renovated, whether through renewal or addition"* (Bayān II, ed. Collins, p. 224, taken from García Gómez 1965, p. 321). From the beginning of his rule, while still emir, in 918–919 (306 H.) *"He also ordered that a pylon be built in the sewage fountain at the entrance of the Alcázar and the gate, called the Gate of Lattice"* (*Ḏikr balād al-Andalus* 1983, [28], p. 126 from trans.).

Years later, in 939, now caliph, "an-Nasir began, after returning from this campaign, to build the attic that was raised on the deposit called "of sin", to the right of the rooftop that leads to the southern gate of as-Sudda, the great one of the Alcázar and which opens on to the avenue. It was projected with battlements and divided into a series of ten gates and, with abundant handiwork, it was soon finished" (Ibn Ḥayyān 1981, Muqtabis V [302], p. 335 from trans.). During his reign, he continued work on the Córdoba palace, preserving the tradition of his ancestors (cfr. Ocaña Jiménez 1935, pp. 165–66).

Of the activities undertaken by Al-Hakam II in the Alcázar, written sources only mention the replacement of the old sābāṭ of Abd Allah because of the expansion that this caliph undertook in the mosque (Pizarro Berengena 2013, p. 240). However, during his reign, work must have been carried out on the interior of the palatial complex, as indicated by the sections of various capitals from different points of the city, but which may refer to their destination: *"for the bedchambers in the Alcázar in the year 353 (=964–65)"* (Ocaña Jiménez 1935, p. 158). Among the activities of the hayib Almanzor to command the power and control the caliph Hišām II, he undertook a reinforcement of the Alcázar: *"he fortified the Alcázar of the Califa . . . with the surrounding wall, he made the pit [jandaq] that ties it at both sides and the gates, protected by guards and night watchmen at his expense"* (Bayān II, ed. Collins, p. 278, taken from García Gómez 1965, p. 334). The *Ḏikr* also referred to these works during the rule of Almanzor: *"All the gates of the Alcázar ex-*

*cept for that of the sayyida were closed by al-Mansur, who imposed order with a strong hand"* (*Ḏikr balād al-Andalus* 1983, [151], p. 189 from trans.). Afterwards, *"When he transferred to al-Zahira [980–981] he was given the title of al-Mansur and ordered that it shall be invoked throughout the minbars of the country following al-Mu'ayyad; he stayed with their slaves in the Alcázar, in which nothing was executed without the order or the consent of al-Mansur. He built a surrounding wall and a pit around the Alcázar of al-Mu'ayyad, he put four night watchmen and guards and installed spies"* (*Ḏikr balād al-Andalus* 1983, [106], p. 192 from trans.)

Regarding the archaeological evidence of these activities of the 10th century in the Alcázar, we summarise the data available up to now:

In the Women's Courtyard, a renewal of the buildings and spaces can be documented from the emir era, period of the Emirate consisting of a brickwork lining, with typical caliphate bonding that alternates a rope with two or three headings, which reinforces and regularises the façade with prior buttresses. The 9th century courtyard was transformed into a corridor, with the building of a new wall parallel to the previous one and with the creation of a new room in its eastern end, which can be accessed through the two bays that, at the same time, were sealed at the end of the caliphate era (Figure 9). To the south of this corridor, a courtyard was prepared that was surrounded by a perimeter platform of calcarenite slabs and a central pavement of marbled limestone slabs. It is likely that this courtyard had perimeter rooms and even a second floor, as shown by the imprints of the stairs attached to the perimeter wall. Regarding this space, there must have been rooms from which some of the gates in the borehole open to the south of the southern façade of the Alcázar de los Reyes Cristianos have been documented (León-Muñoz 2020a, pp. 295–98).

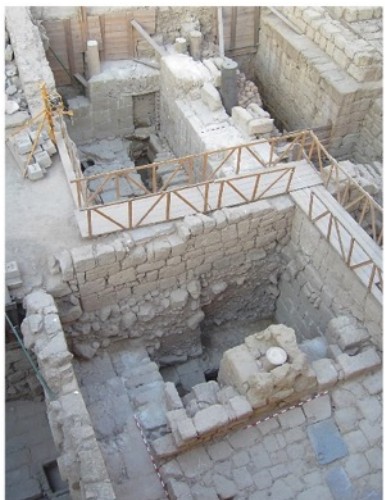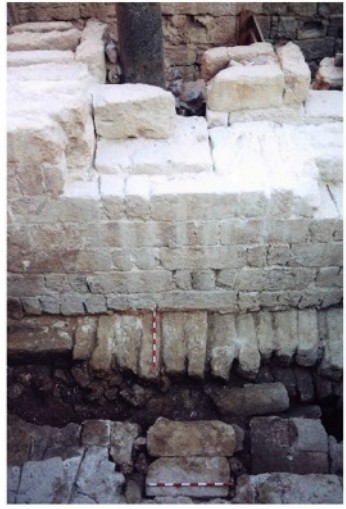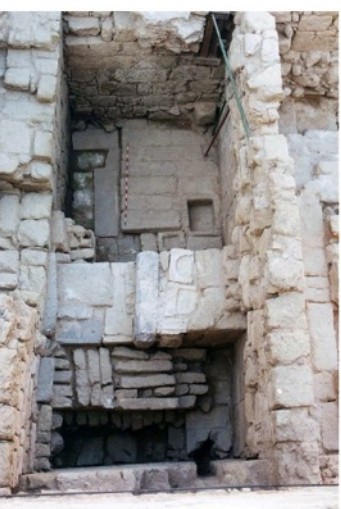

**Figure 9.** Reforms of the caliphal period in the Patio de Mujeres del Alcázar de los Reyes Cristianos (Women's Courtyard of the Christian Monarcs' Fortress) (image: GMU-UCO Agreement).

Regarding the South Courtyard of the Episcopal Palace, two great constructive phases of the caliphate era have been identified. The first of these, documented in the most easterly zone, is related to the construction of the *sābāṭ* of Al-Hakam II and consists of a strong wall parallel to the eastern façade of the Alcázar, made with large brickwork and careful stereotomy, which supported the bridge in the interior of the palace (Figure 10). To the west of this wall, there is an extensive room, which is accessed from the west side, with walls painted with red oxide clay and paved with marble slabs on two levels, destined for a latrine holding a cleaning tank. The second phase consisted of the construction of a corridor that runs north–south, set immediately to the east of the wall that supports the sābāṭ, made with walls of brickwork that are much thinner, paved with lime mortar soil. These structures appear to be related to the guard corps that controlled access to the private part of the Alcázar (Ortiz Urbano 2022, p. 181).

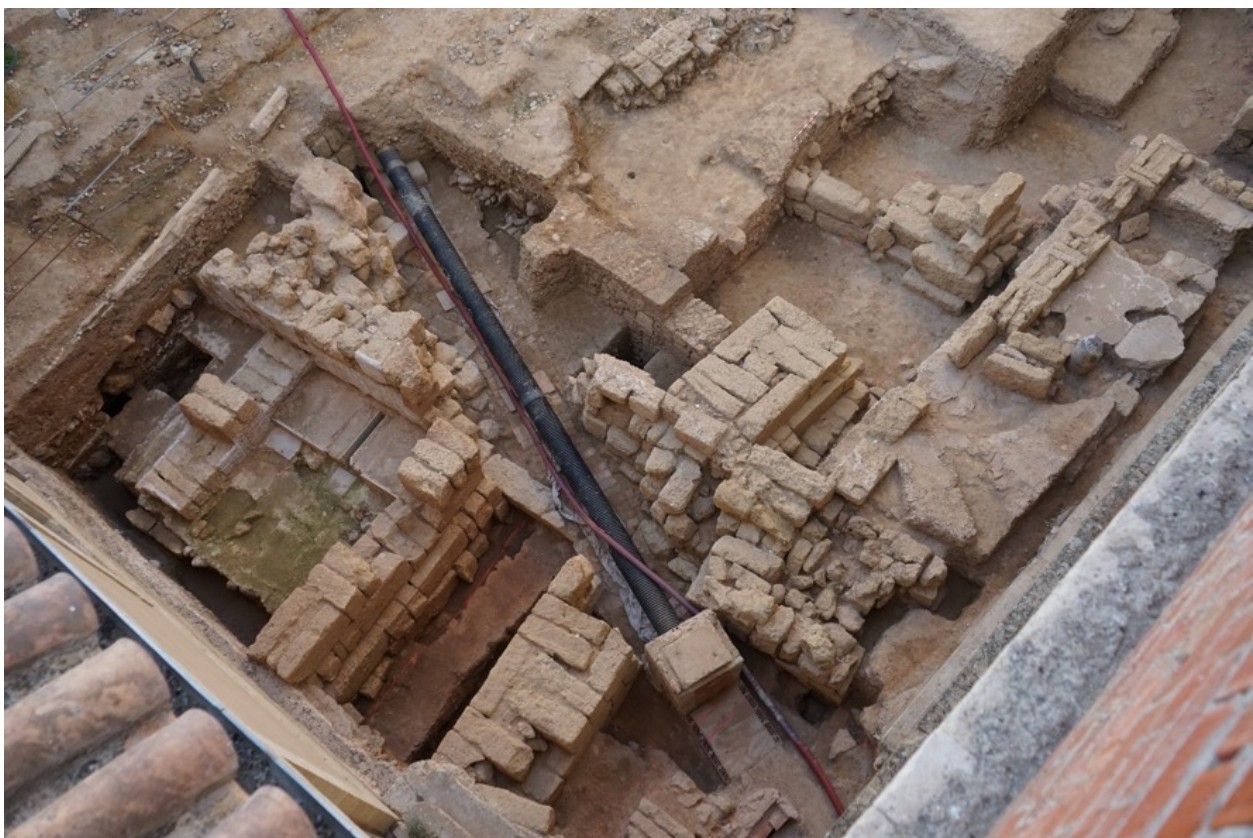

**Figure 10.** Aerial view of the intervention in the Patio Sur del Palacio Episcopal (South Courtyard of the Episcopal Palace). Structures associated with the sābāṭ made with an ashlar of great size and careful stereotomy can be distinguished; the latrine at its left; the corridor that could have held the guard corps defined by ashlar walls with a much narrowed module, paved with lime mortar floors (image: Raimundo Ortiz Urbano).

As part of the preventive archaeological activity undertaken in the grounds of the current "Parking La Mezquita", the north wall of the Umayyad Alcázar was documented, dated to the end of the 10th century, and which could be linked, as such, to the fortification work performed by Almanzor. This building consisted of a strong wall made entirely from calcarenite brickwork, with a specific zig-zag layout that appears to adapt its layout to the existence of the caliphate baths located to the south (Vargas Cantos et al. 2010, pp. 381–82). This stretch of wall accounted for a space designed for a guard corps that controlled the entry to the gate and was documented in this northern wall section as the *Bāb al-Hammam* (Murillo Redondo and León-Muñoz 2019, p. 135) (Figure 11).

As a matter of fact, the main construction identified in the interior of the Alcázar corresponds to the *hammam* excavated in the north sector of the Plaza Campo Santo de los Mártires. The main part of the structures discovered in the subsequent excavations correspond to the caliphate phase, forming typical rooms in this kind of building: a changing room, a latrine, and the succession in a right angle of the three rooms (cold, mild, and hot) with the oven and the boiler[8] (Marfil Ruiz 2004, p. 59; Murillo Redondo and León-Muñoz 2019, pp. 136–37, 2014). These structures have basically been dated from some elements of architectural decoration recovered in the bathtub such as "*a sealed roof bow of gypsum of three lobes and springlines of others, voussoired on small columns. Its style corresponds to al-Hakam II or an era just a little later*" (Torres Balbás 1957, p. 618).

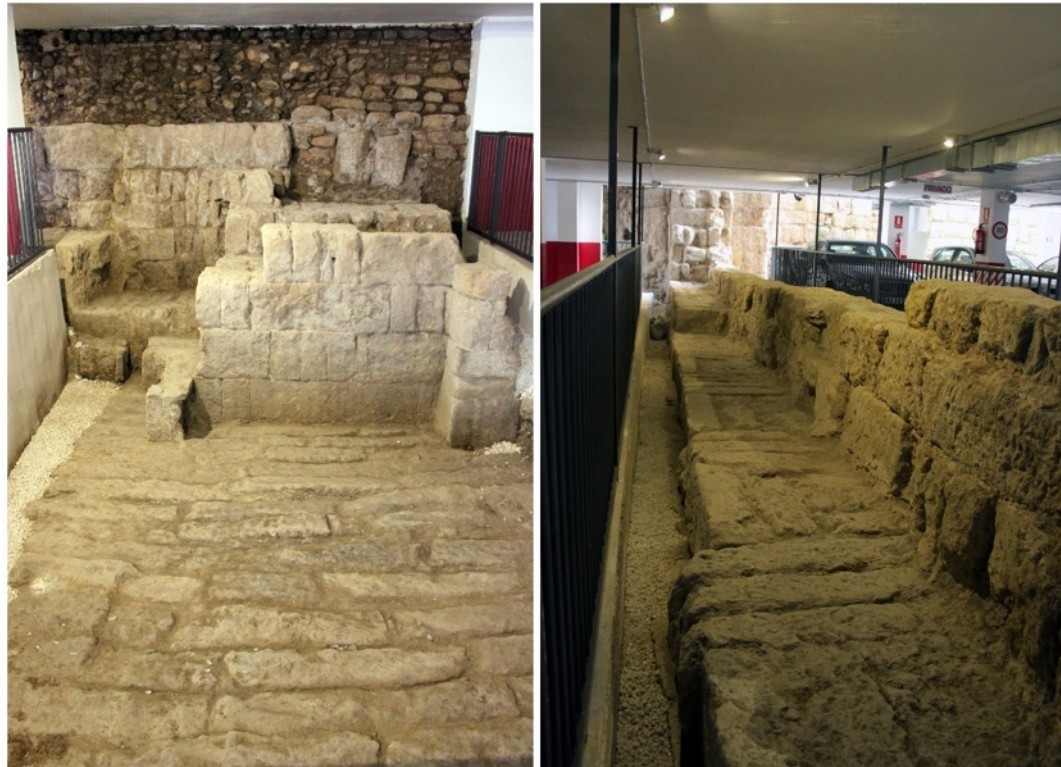

**Figure 11.** North wall of the Umayyad Alcázar (Fortress) documented on the location of the current "Parking La Mezquita" (Image Alberto León).

Since the beginning of the 11th century, the Alcázar has held a prominent place in written sources as the stage for eventful episodes related to the outbreak of *fitna*, and by being the building held by the subsequent heirs to the throne. However, the evidence news does not mention any new building, but the pillaging of some of these rooms (García Gómez 1947, pp. 284–85). Notwithstanding, during the 11th century, the remodelling of the previously mentioned *hammam* was performed, which added a large porticoed room to its eastern side, with two side bedrooms that have been understood to be a reception room erected during the occupation of Córdoba by the Seville Abad *taifa* (Marfil Ruiz 2004, p. 63). So far we have presented the available archaeological information corresponding to the Caliphate period of the Alcázar. The documented remains do not allow a clear identification of the spaces mentioned in the written sources (except those related to the sābāṭ), but they do give us an idea of the entity of the constructions, the reuse of previous structures and the dense occupation of the spaces available in inside this walled enclosure.

*4.3. The Alcázar and the Almohad Alcazaba*

The great transformation of the Umayyad palatial complex was performed during the Almohad era, specifically in the historical period that, until recently with the research on Mādinat Qurṭuba, has been the least addressed. The old Umayyad Alcázar (*qaṣr Qurṭuba al-ʿatīq*) is briefly mentioned on a few occasions by the scribe Ibn Ṣāḥib al-Ṣalāt regarding each room of the caliph Abū Yaʿqub Yūsuf in Córdoba. The first of these, in August 1171, mentions a Maŷlis al-Yumn (Room of Happiness)[9], as a reception where the sovereign received the welcome of the prominent people of the city. This space is not mentioned in the descriptions of the Umayyad building, so the possibility of it being a later construction has been suggested (Zanón 1989, p. 77); it could otherwise suggest a name change of a pre-existing space. The second reference, in June 1172, only indicates that the caliph was

in the Alcázar for seven days as he prepared for a punitive campaign against the city of Huete (Ibid., pp. 76–77; Ibn Ṣāḥib al-Ṣalāt 1969, *Al-Mann bi l-imāma*, p. 205 from trans.).

The only explicit documentary information on the construction of palaces[10] in Córdoba during this era refers to the Qaṣr Abī Yaḥyà, erected by the son of the caliph Abū Yaʿqub Yūsuf. According to news transferred by ʿIbn IḎārī, the emir Abū Yaʿqub al-Manṣūr (1184–1199), brother of Abū Yaḥyà, stayed in this palace in the summer of 1190 (586 H.) (Zanón 1989, p. 80; Huici Miranda [1956] 2000, p. 346). The dates indicated for the two references to the Alcázar and this new palace provide a chronological interval for the construction of this *Qaṣr Abī Yaḥyà*. Al-Maqqari alludes to this palace from the Almohad era (Nafh al-tib, I, p. 470, taken from Zanón 1989, p. 80). From this information, it has been deduced that "*resting with arches on the Guadalquivir, on whose construction, according to the chronicle called "Anonymous of Copenhagen", great amount were spent*" (Torres Balbás 1949, p. 30). From these brief textual references, Torres Balbás saw as belonging to this building "*some brick walls, opened with balconies of acute horseshoe arches, existing on one of the waterwheels of the Guadalquivir, close to the bridge*" (*Ibidem*). In recent work regarding Almohad Córdoba, this old hypothesis, which intends to identify this palace with the waterwheel of Albolafia, has been reconsidered, which, for R. Blanco, "*its exact location has still not been confirmed archaeologically*" (Blanco Guzmán 2019, p. 48, 2022, p. 107). According to our criteria, this affirmation entails omitting all of the eloquent archaeological information recovered in recent years in this urban sector.

The excavation of the Women's Courtyard and the reading of the paraments of the Alcázar de los Reyes Cristianos has enabled the documentation of a phase of the Almohad era that shows the transformation of the structures of the old Umayyad Alcázar into an Almohad palace (Figure 12). For the construction of this building, part of the structures of the Umayyad era were torn down and the land was regularised with a strong filling packet that raised the elevation of the soil by 2.5 m (León-Muñoz 2013; León-Muñoz 2020a, p. 299). A building was erected on this surface, apparently separate from the rest of the palatial complex with a square floor, made by transporting ashlar from the previous structure and organising it into two large sections. The eastern part, destined for the service rooms, was rounded by a great sewer whose layout marks the main axis of the structures and coincides with the original entry door, which still preserves imprints of the springline of the horseshoe arch, covered by the vaults of the baroque entry (León-Muñoz 2013, p. 342; León-Muñoz 2020a, p. 303; Murillo Redondo 2020, p. 239). The western half, dedicated to the court area, was occupied by a rectangular cross-platform courtyard, with reduced garden beds and porticoed rooms on their lesser sides, covered with side *alhanias* (Murillo Redondo and León-Muñoz 2019, p. 149; Murillo Redondo 2020, p. 242). The prototype architectural models for this type of courtyard are found in the Alcázar of Seville from the Almohad era (Rodríguez Moreno 2011, pp. 193–94). After the excavation and later restoration performed in the 1950s by the municipal architect, Víctor Escribano, this area became known by the confusing term of "Morisco" or "Mudéjar Courtyard" (Escribano Ucelay 1972). In the north-east square, this courtyard would connect to a private bathroom, identified by Escribano as the "Bathroom of Doña Leonor" and attributed to the era of Alfonso XI, but which preserves traces of previous phases that we connect to the Almohad palace (Murillo Redondo and León-Muñoz 2019, p. 147).

Regarding the Qaṣr Abī Yaḥyà, Al-Maqqari provides a revealing anecdote: "The sayyid was asked: How have you endeavoured to build this Alcázar by separating yourself from the population of Córdoba? And he said: I know that they will not remember a wālī upon being dismissed, nor will it mean anything to them because the Marwan caliphate continues in their memories. I have wanted to leave a print of my own in this land which shall be remembered in spite of it" (Nafh al-tib, I, p. 470, taken from Zanón 1989, pp. 80–81). Some interesting considerations may be extracted from this text. On the one hand, the demolition of Umayyad structures, on which a new building is erected, is a clear example of the propaganda message of the Almohad domain, which intended to erase or replace the work of the Marwan caliphate. This same procedure has been documented in the Almohad

Alcázar of Seville, where the work involved the demolition of part of the previous taifa structures and the filling and levelling of the paving elevations (Tabales Rodríguez 2010).

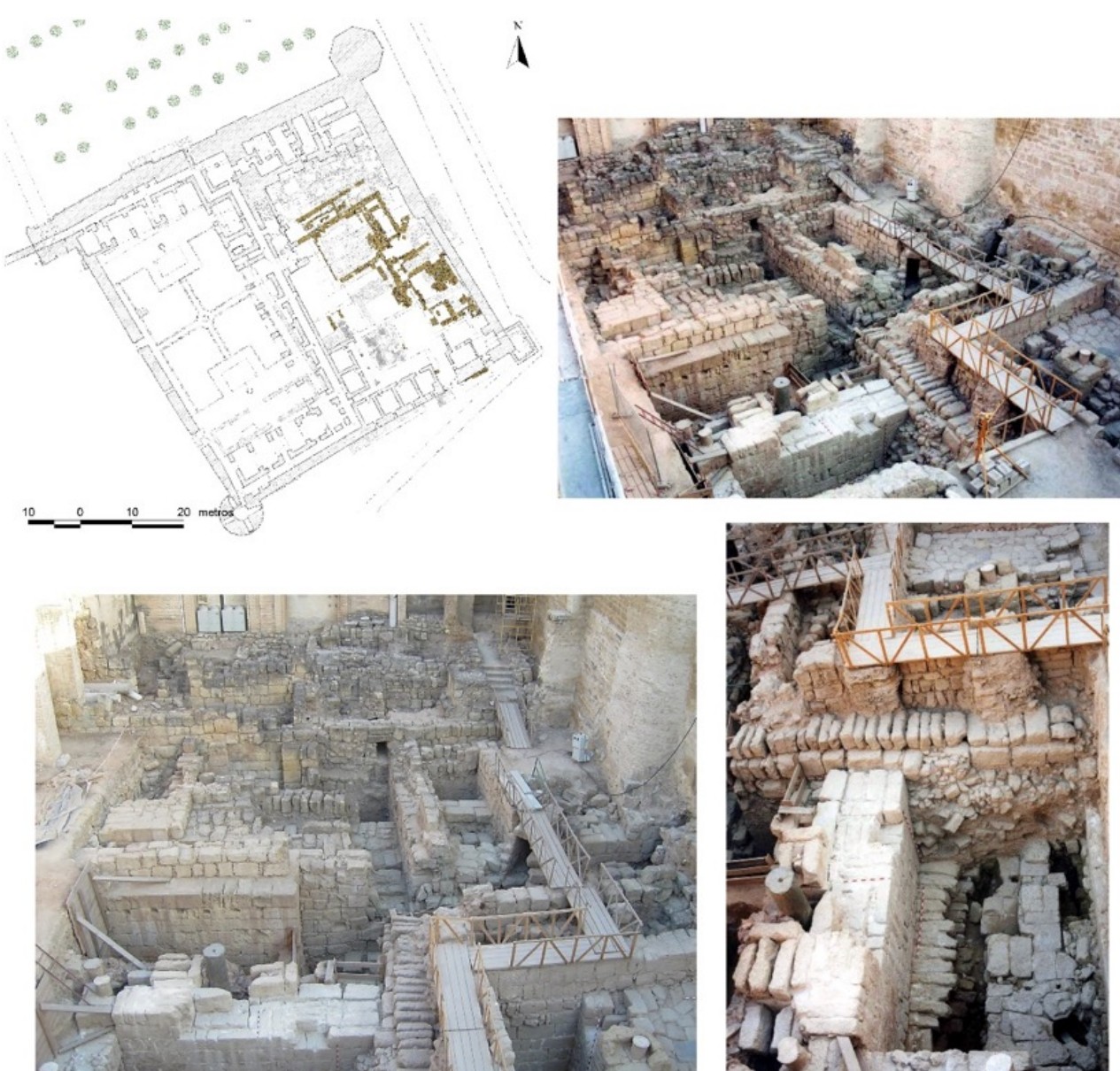

**Figure 12.** Floor and general view of the structures documented on the Patio de Mujeres del Alcázar de los Reyes Cristianos (Women's Courtyard of the Christian Monarcs' Fortress), belonging to the Almohad period palace (image: GMU-UCO Agreement).

This palace is, according to our criteria, referred to in the *Latin Chronic of the Kings of Castile*, which tells of the first activities of King Fernando III on 30 June 1236, after the conquest of the city of Córdoba. Holding mass in the recently consecrated Cathedral of Santa María: "*the king entered in the noble palace which the kings of the Moors had prepared, of which great things were said by those who had visited and those who had not visited deemed incredible*" (Escobar Camacho 2020, p. 384). However, Escobar indicates that its exact location within the Al-Andalus Alcázar is not known and proposes that the Royal Alcázar or the King's House would have been located in the location of the current Seminary of Saint Pelagius: "*This royal Alcázar* [which would possibly coincide with the palace mentioned by the scribes as the place where Fernando III arrived on the 30th of June 1236], *which was in*

*use during the 13th century and a good part of the following, is documented as houses of the king, royal Alcázar (Alcázar real o real Alcázar)"* (Escobar Camacho 2020, p. 389).

This building would form, undoubtedly, the palatial nucleus of the new *alcazaba* extending to the west and the south of the walled Umayyad area (Figure 13). The strategic importance of Córdoba, in a communications crossroad with the major bridge as a key piece, was reinforced in this era of intense military activity with the stationary movement of troops, who concentrated, generally, in the capital of the Guadalquivir Valley. The result was the reinforcement of the city defences on both sides of the river with the construction of fortified grounds on each side, which have been dated to the beginning of the 1170s (cfr. León-Muñoz et al. 2008; León-Muñoz and Murillo Redondo 2009, p. 427). On the enclave occupied by the Umayyad Alcázar, an intense and ambitious architectural program was conducted that involved the renewal of the defensive elements and the notable expansion of its perimeter with the addition of an extensive *alcazaba* that corresponds to the walls of the now-called "Old Castle of the Jewish Quarter" (León-Muñoz 2013; Murillo Redondo 2020, pp. 247–48).

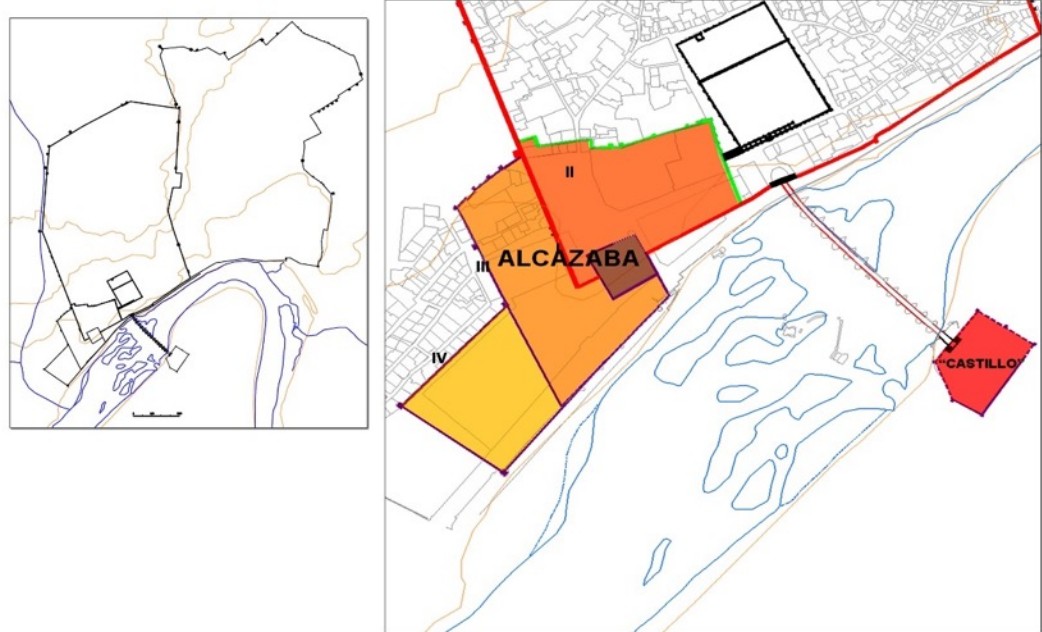

**Figure 13.** Reconstruction of the floor of the new Almohad citadel of Córdoba, which extended towards the west and south of the Umayyad walled enclosure (image: GMU-UCO Agreement). The green line marks the perimeter of the caliphal fortress. The purple lines mark the extensions from the Almohad period. And the space filled in brown indicates the location of the Almohad palace.

Inside the old Alcázar, new constructions were performed such as the structures documented in the south courtyard of the Episcopal Palace; at the same time, the annulment or sealing of some of the accesses was conducted such as the called "Gate to the Baths" in the northern face of the area (Murillo Redondo 2020, p. 247). At the same time, some of the facilities were remodelled such as the great *hammam* of the Campo Santo de los Mártires, which had a new bath added in order to replace the old caliphate one and was accessed through the *taifa* reception room. This phase was previously documented in the interventions of Félix Hernández from the architectural decoration of the chiselled plaster recovered (Ocaña Jiménez 1975); information confirmed by the latest excavations (Marfil Ruiz 2004, p. 64).

Definitively, the look of this complex at the moment of the Castilian conquest of Córdoba must have been far from that where the first Umayyad rulers and emirs were installed, the fruit result of eight centuries of intense building activity promoted by the political power as a projection of its authority.

## 5. The Umayyad Alcázar: About the Whole and Its Parts

### 5.1. The Limits of the Al-Andalus Alcázar

As we have previously stated, one of the topics that the research has primarily focused on is related to the limits of the palatial complex, greatly conditioned by the reading of the textual documentation and, to a lesser extent, by the material evidence. The information provided by the Al-Andalus authors provides dimensions that have been transferred onto the plan with different proposals. Thus, Al-Maqqari provides a figure of 1100 cubits in outline (=517 perimeter metres) (García Gómez 1965, p. 322). Regarding this, the information provided by Al-Ḥimyarī is revealing, according to which "*the Alcázar is to the west of the city; the occupied lands extend to the southern and western walls*" (Arjona Castro 1982, doc. 298, p. 233; Torres Balbás 1957, p. 590).

Beyond these textual references, the archaeological evidence available allows for a more or less exact approximation of the limits and distribution of the Umayyad Alcázar. In this sense, the proposal made at the end of the 20th century by Montejo and Garriguet provided quite a precise plan for the limits of the building from the archaeological records known until then (Montejo Córdoba and Garriguet Mata 1998, pp. 325–26). The interventions performed in the Women's Courtyard of the Alcázar de los Reyes Cristianos have completed this general form, with the inclusion of a prior area in the south-western sector of the city that should be included in the palatial complex (León-Muñoz and Murillo Redondo 2009). In its northern extreme, the zig-zag layout of the northern wall also lightly modified the regular layout of the area. The result is a complex with a slightly more irregular aspect, but based on real material evidence.

This being the case, with these new data, the limits of the area would be outlined, to the east, by the façade of the episcopal palace, bordering the Congregational Mosque, which still preserves the northern half of its layout and various metres of elevation (Castejón y Martínez de Arizala 1929, p. 27; Montejo Córdoba and Garriguet Mata 1998; Marfil Ruiz 2010). These walls were extended to the south until they connected with the southern wall of the city, as has been documented by the excavation of the South Courtyard of the Episcopal Palace (Ortiz Urbano 2022). The northern façade also preserves a good part of its layout, of which various towers are visible such as the famous courtyard of the Córdoba Conference Hall and, more recently, those documented during the restoration of the north section (León-Muñoz 2020b). From this point, the north façade extended to the west, where a section is currently visible in the courtyard of the Provincial Library, known as the Bishop's Gardens, excavated by Félix Hernández and Ana Mª Vicent at the beginning of the 1970s. At the western limit, where it would have connected to the urban wall, the wall has a unique zig-zag layout towards the north, probably to adapt to the caliphate *hammam* located in the Campo Santo de los Mártires (Vargas Cantos et al. 2010; Murillo Redondo and León-Muñoz 2019). From this point of view, the western limit of the Alcázar appears to coincide with the wall of the Medina, which descends towards the south, fossilised in the façade of the residences that mark the western closure of the Santos Mártires Square and its elevation was reused for the construction of the eastern wall of the Royal Stables (Pizarro Berengena 2008, p. 1719; Lara Jiménez 2020, p. 450). The south-western sector of the Alcázar, where it would have connected to the wall at the south of the city, is not currently visible, but appears to be fossilised in the marked change in the elevation outlined in the gardens of the Alcázar de los Reyes Cristianos and appears represented as such in the drawing by Guesdon from ca. 1863. Finally, the Alcázar's southern layout includes the walled area (*castellum*) of the Late Antiquity era (*cfr.* Murillo Redondo and León-Muñoz 2019), to the east of which its layout would coincide once again with the urban wall.

With the information currently available, we can establish quite an approximate calculation of the area occupied by the Umayyad Alcázar at the end of the 10th century, which covered a surface area of around 42,720 m$^2$ and had 944 m of external perimeter wall. However, this area notably increased after its transformation into an extensive Almohad alcazaba around the final quarter of the 12th century, with the cre-

ation of various walled areas (such as the known "Old Castle of the Jewish Quarter") (León-Muñoz 2013; Murillo Redondo 2020), reaching up to a surface area of 102,695 m$^2$ and 1520 m of walled perimeter.

However, recently, Alberto Montejo proposed a different reading for the limits of the palatial area, which considered the archaeological discoveries documented in the current Alcázar Viejo[11] neighbourhood to the west of the Medina and the Alcázar itself. In these interventions, great architectural structures have been documented, with brickwork walls and calcarenite slab paving and lime mortar painted with red oxide clay. According to Montejo, "*the results obtained in the excavations of four grounds in the Alcázar Viejo neighbourhood, inside the Royal Stables and the news of old discoveries, lead to us today suggesting that the Umayyad Alcázar of Córdoba reached west and, as such, would occupy an area far greater than we thought*" (Montejo Córdoba 2015, p. 5).

The archaeological evidence is speaking of, no doubt, high level constructions, linked to the circles of Umayyad power. However, they might not necessarily form part of the walled area of the Alcázar, but they must have formed part of the buildings in the immediate surroundings. This type of unique construction has been documented in the surrounding area of the Congregational Mosque or to the north of the Alcázar itself, in C/Manríquez or C/Tejón y Marín, understood as residences for people of a high social level (León-Muñoz and Montejo Córdoba 2023, pp. 209–10) or facilities linked to the administration of all Al-Andalus.

As the seat of the Umayyad government, it must be considered whether the Alcázar of Córdoba could have held the buildings and institutions that formed the bodies of political and administrative functioning of the emerging Al-Andalus state. The Alcázar of Madīnat al-Zahrā', despite the difficulties of its exact delimitation, occupied a total surface of 19 ha including the garden spaces, of which 10 ha have been excavated, corresponding to the central part of the upper platform (Vallejo Triano 2010, pp. 222–23 and Figure 8). With this area, it is logical to think that its Alcázar could have held all the institutional apparatus. Against these proportions, the Alcázar of Córdoba occupied an area of just 4.2 ha, so it is apparently evident that the palatial complex availed of a limited space and that this would have been used to the maximum (Figure 14). A sign of the density of structures is the information transferred in the intervention of 1928 in the Campo de los Mártires, where "*a series of rooms have been cut, with their walls and paving, which indicate that all of it was already built*" (Castejón y Martínez de Arizala 1928, p. 34). Perhaps because of this, the tradition of assigning palaces and properties outside of the Alcázar to the sons of emirs and caliphs once they were of legal age was established (except for heirs, who remained in the palace) in which they could enjoy all the comforts and occupy them with their family members and an extensive staff. The chronicle of Abd Al-Rahman III, who followed the customs of his predecessors (Ibn Ḥayyān 1981, *Muqtabis V*, [7–8], pp. 20–22 from trans.), expressly mentions it. Nor was it the residence of viziers and chamberlains (as occurs in the cities of Al-Zahra and Al-Zahira), who had their houses outside the palace, which they attended to perform their functions, as is the case of Abdalkarim (Ibn Ḥayyān 2001, *Muqtabis II*, 1, [117v], p. 97 from trans.). This being the case, it seems convenient to enrich the image of the Alcázar with extensive gardens, whose presence and extension must certainly have been limited. According to the names, they may be located in the areas of Bāb al-Ŷinan (Gate of the Gardens) and the Rawḍa (or Funeral Garden). There must have been even more courtyards (sāḥāt) lined with porticos (portals) as ante-rooms for some of the main living rooms.

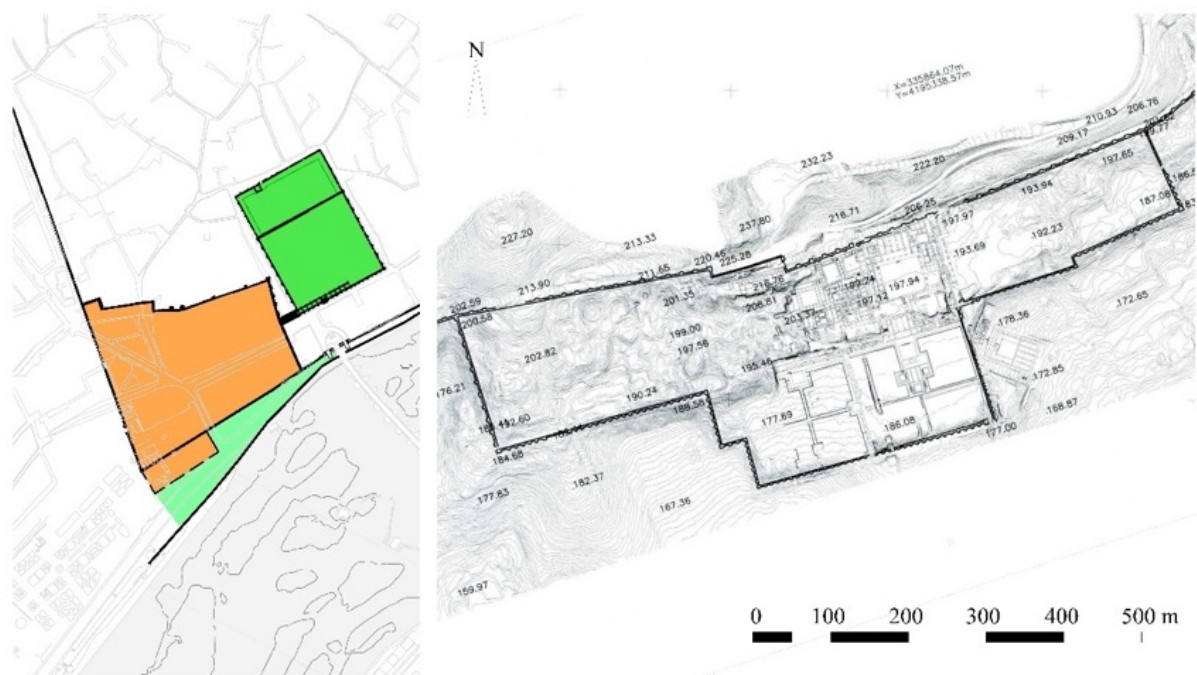

**Figure 14.** Comparison at the same scale between the extension of the fortresses of Córdoba (**left**) and Madīnat al-Zahrā' (**right**). Own elaboration from León-Muñoz and Murillo Redondo (2009) and Vallejo Triano 2010. The orange color indicates the surface occupied by the Umayyad fortress of Córdoba. The green color marks the aljama mosque.

*5.2. Internal Layout*

Even though the limits of the complex are more or less well-defined, the level of knowledge on the internal layout of the spaces and the characteristics of the buildings that formed it are still only partially known. The result of the constructive process was described at the end of the 10th century as a complex of palatal pavilions or rooms (called *maŷlis, dār, bayt*), some of which appeared in the references of Al-Maqqari, who took them, at the same time, from Ibn Baškuwāl[12]. The written sources and the material evidence provide parallel information that only coincide on exceptional occasions by unequivocally having recognisable elements.

The few archaeological excavations undertaken in the interior of this expansive area have partially documented some structures, but it is not possible to link them to any of the pavilions mentioned in the chronicles. For the moment, we can only make a general characterisation of the complex while waiting for the completion of programmed activities that try to resolve the numerous questions that this architectural complex still provokes.

Because of this, the priority in the current state of research should not be the physical identification of the spaces of the Alcázar, thus avoiding falling into the recurrent philological archaeology, but instead into the definitions of its key points, the stage and buildings in which the power that it holds is shown. There are two main activities that state the legitimacy of the governors and that is explicitly and frequently repeated by the sources (specifically, in the *Ḏikr bilad al-Andalus*): the occupation of the Alcázar by the governor, and the burial in the funeral garden of the interior of the palatial complex[13]. We are going to expose, basically from the textual information, the specific references to three of these spaces of representation of power in the Alcázar: the doors, in particular, the Bāb al-Sudda, the throne room and the Rawḍa or funerary garden.

The Alcázar is, above all, the seat of civil authority, and as such, "the Qaṣr legitimises whoever occupies it" (Viguera Molins 2020, p. 127)[14]. It is the point of reference to which all of the new governors turn when trying to confirm their sovereignty. This is what the founder of the Andalusian Arabian Umayyad dynasty, 'Abd al-Raḥmān al-Dājil, did

after defeating his enemy, Yūsuf al-Fihrī, when "*he entered the Alcázar, thus achieving total power, the utmost glory and general submission*" (*Ḏikr balād al-Andalus* 1983, p. 121, trans. from Viguera Molins 2020, p. 128). Like him, the other Umayyad emirs and caliphs who ruled Al-Andalus "occupied the Alcázar of Córdoba" as a sign of their dynastic legitimacy (*Ḏikr balād al-Andalus* 1983, [89], p. 115 from trans.). However, after the *fitna*, the rulers of the *taifa* of Córdoba abandoned this custom. This is how Abu l-Hazm b. Yahwar acted, who during the twelve years that his rule lasted "*did not enter the Alcázar, he was not called sovereign nor was he given any honorific title and he was not allowed to be invoked in sermons nor have his name engraved on coins*" (Ibn Iḏārī 1993, *La caída del Califato* ["The fall of the Caliphate"] [3], p. 227 from the trans.). Even after his death, he was buried in his house, and not in the Umayyad funeral garden (*Ibid*., p. 228).

The main stages where the authority of the sovereign is declared in the most evident manner are, specifically, its doors. Because of this, they are the most frequently mentioned elements in the sources, unlike other spaces found in the building. Behind them, encounters between the governor and the population caused by events of special significance occurred; this is where the prestige of sovereignty is evident[15]; together with these, some of the authority's functions are exercised (Finster 2006, p. 359). Therefore, the deep symbolic load that they bear is shown in its structure, in the architectural elements that form them, and that are used once more by them.

Of these doors, this power is represented most clearly and completely by the Bāb al-Sudda, "the largest one in the palace" (Ibn Ḥayyān 1981, *Muqtabis V* [301], p. 333 from trans.), explaining its abundant presence in the chronicle sources. Its name became a "*symbol of the chancery and even of all the caliphate power*" (García Gómez 1965, p. 326) and, as such, it is a term that covers both the seat of power, "an Alcázar in whole", and the court itself (Viguera Molins 2020, pp. 142–43). The existence of a Bāb al-Sudda in Madīnat al-Zahrā', identified as the central door of the great monumental portico for entry to the official area of the Alcázar (Vallejo Triano 2010, p. 227), states the same idea in the end.

It is a place mentioned in ceremonious episodes, in front of which the sovereign applied justice and punishment for rebels who threatened or betrayed this power; retinues would leave from there and praise was given before military expeditions led by the emir or caliph; in the room built on the upper floor (the roof terrace or *as-saṭḥ*), the heir "was enclosed" during the caliph's absence; the official rooms (eastern room, like in the Medina Azahara) were entered through them; the *zalmedina's* chair was located in front of them and there he exercised his jurisdiction (Ibn Ḥayyān 2001, *Muqtabis* II-1, p. 91); from the roof terrace, the caliph handed out alms to his poorest subjects, etc. We know with certainty its location in the south-eastern façade of the Alcázar[16], and that it was subject to architectural interventions by 'Abd al-Raḥmān II[17]. However, we know practically nothing of its architectural characteristics, only that "*The Gate which has a roof terrace above it has a projecting balcony, which is unrivalled in the world*" (Al-Maqqarī 1855–60, p. 303, taken from Arjona Castro 1982, doc. no. 273, p. 207). According to that preserved in other monumental portals from the emir period, it is logical to think that the main door of the Alcázar corresponds to the same architectural models of contemporary buildings, where the most relevant examples are found in the Umayyad Congregational Mosque. The Door of the Viziers was remodelled in 833 by 'Abd al-Raḥmān II including an eave projecting over the modillions of rolls in the upper part of the central arch (Márquez Bueno et al. 2021, p. 23). Perhaps this eave could be a modest imitation of the one that supported the projecting balcony of Bāb al-Sudda mentioned by Ibn Ḥayyān.

As we have already indicated, the main door of the Alcázar had great symbolic meaning. Similar to what happens in many palatial complexes in the East, the doors served to display military triumphs by reusing conquered doors or their sheets (Finster 2006, p. 360). This same procedure was followed in the Bāb al-Sudda of the Umayyad Alcázar of Córdoba, whose doors were "*covered with iron plates held in place by an artistically-designed copper band, which represents a figure of a man with an open mouth. This extraordinary work of art, which is found on the lowest part of the door, served at the same time as a knocker and belonged to one of*

*the doors of Narbonne in the land of the Franks. The emir Muhammad brought it here when he conquered that city: he took it from there and brought it to Córdoba*" (Al-Maqqari, *Analectes* I, p. 303, taken from Arjona Castro 1982, doc. no. 273, pp. 207–8).

The other form of officially legitimising power in the Alcázar is by taking possession of the throne, where the sovereign receives an oath of loyalty, or his presence in the ceremonious rooms, presiding over the ceremonial acts, in which he receives the State's civil servants. In this regard, various pavilions are mentioned in which these receptions were performed. One such example is 'Abd al-Raḥmān III in the year 912 "*He sat on the throne to receive an oath of loyalty from his subjects on the 1st Thursday of the mentioned month of Rabi I (=15th of October) in the Maŷlis al-Kāmil of Córdoba (. . .) The oath ceremony ended (. . .) 'Abd al-Raḥmān left the throne to perform the funeral prayer for his grandfather and bury him in the grave of the Rawḍat al-julafā'*" (*Una crónica anónima de 'Abd al-Raḥmān III an-Nāṣir* (1950, [2], pp. 91–92)). This room already existed in the middle of the 9th century, called Dār al-Kāmil, mentioned by Ibn al-Qutiyya upon the death of 'Abd al-Raḥmān II (Ocaña Jiménez 1935, p. 165).

During the eventful episodes of the *fitna*, a throne room in the Alcázar of Córdoba is mentioned on two occasions. The first of these takes place in 1010 during the usurping of Muḥammad [II] al-Mahdī (1010): "*Once again in Córdoba, al-Mahdī took over the Alcázar and had oaths sworn, after which he called upon Hišām al-Muʿayyad, he sat him at his side and requested that he abdicate in his favour; Hišām did so, putting it in writing and certifying it. The fatá Wāḍiḥ, who was present, angered by all this, left, gathered all the Amirite fatás and led them to the Alcázar while shouting: We do not obey Hišām al-Muʿayyad. They entered the Alcázar, they took al-Muʿayyad, they sat him in the throne and invoked his motto. Ibn 'Abd al-Ŷabbār, who was in the bath, was removed and brought before Hišām, who was in the Maŷlis al-Jilāfa with the fatás before him*" (*Ḏikr balād al-Andalus* 1983, [166], pp. 210–11 from trans.).

The second event, which took place in 1024, is related to the enthronement of Muḥammad [III] b. 'Abd al-Raḥmān, who "*they took to the royal palace* (dār al-mulk) *which was deserted, they sat him down in shock in the midday room*" and there they named him caliph (Ibn Iḍārī 1993, *La caída del Califato*, [139], pp. 122–23 from the trans.). It could be that it refers to the same building, according to the likely location of Dār al-Kāmil in the immediate surroundings of Bāb al-Sudda, in the southern area of the Alcázar.

There are other pavilions of the Alcázar that also held ceremonious receptions such as the Western Room (*maŷlis garbī*) of Dār al-Rawḍa, where the Fast-Breaking Party was held in 975. There, Al-Hakam II "*sat in his throne*", received "*the Brothers, the viziers and the upper palace civil servants*". In the same celebration, his son, the future Hišām II, did the same with everyone in "*the al-Zahrā room*' (maŷlis al-Zahrā'), *at Ḥāʾir, with the most complete and perfect protocol*" (García Gómez 1967, [237], pp. 271–272 of the trans.)[18]. These upper civil servants, before being received by the caliph, stayed, according to their rank, in the portal (pórtico) of Dār al-Kāmil (the viziers); at Bayt al-wizāra (the Qurays and the Banu Umayya); at Balāṭ al-Rīḥ (the māwlas); in the tribune of this balāṭ (faqihs and qadis); in the bedchambers of the palace civil servants (the army members) (García Gómez 1967, [237], p. 273 from trans.; García Gómez 1965, p. 332).

As can be deduced from this information, in the surroundings of these reception pavilions, there were other rooms aimed to hold some of the administrative bodies of the State. The oldest one which we have records of was prepared by 'Abd al-Raḥmān: "*He was the first to introduce the daily appearance of the viziers in the Caliphate Alcázar, so as to speak to them about the matters of the kingdom that he wanted to and to ask them for their opinion regarding those, whether individual or collectively, as he also provided for them in his Alcázar with a good lodging designed for meetings and sessions, whose use continues until today under the name of "Ministerial House" (Bayt alwizārah), from where he called them to his dais. . .*" (Ibn Ḥayyān 2001, *Muqtabis* II, 1, [144r], p. 183–84 from trans.). Another of these meeting rooms aimed at administrative and legal functions is the "Elm Room" of the Alcázar, where the faqihs of the city were called (Ibid. [175v], p. 279 from trans.).

As we have mentioned, the third stage in which the dynastic legitimacy of the occupants of the Alcázar of Córdoba is displayed is the Rawḍa (Rawdat al-Julafa)[19]. The custom of burying in the funeral garden inside the palace, alongside the remaining members of the Umayyad dynasty who ruled Al-Andalus, grants an "*important and palpable link to the past and tradition and, as such, a source of legitimacy*" (Montejo Córdoba 2006, p. 239). Although some proposals place the royal cemetery outside the defined limits of the Alcázar[20], the most solid hypothesis locates this space in the surroundings of the southern façade of the palace; specifically, in the space occupied by the Seminary of Saint Pelagius. This interpretation is based on a complex of architectural and decorative materials recovered during the work to expand the seminary and were donated to the National Archaeological Museum in 1868. Standing out among these is a fragment of white marble with a Kufic inscription translated as "... on her ... with her ...". Using this formula, this piece has been understood as a funeral entry, dated from the second half of the 10th century, dedicated to a female character (Montejo Córdoba 2006, p. 252). However, recently, a different chronological and functional interpretation has been proposed for this fragmentary inscription: it has been dated to the middle of the 9th century and "*certainly celebrated the building of a place designed for performing ritual ablutions*" (Barceló 2018, p. 9). The location of this funeral space in the southern face explains the construction of the waterwheel (the Albolafia) in the period of the Emirate at the foot of the Alcázar[21], in order to water the gardens that give it its name.

Regarding the activities of the court environment (music, poetry, discussions, drinks, etc.), which the Umayyad sovereigns were very fond of and are mentioned repeatedly in the sources, it is logical to think that they should have been developed in private or semi-private spaces; that is to say, other than in the buildings of official reception (cfr. Anderson 2018, p. 222). Some of the pavilions mentioned by Ibn Baškuwāl held these activities such as the Al-Mubārak Room (The Blessed). In this room, the musician Ziryāb organised "*a joyous meeting*" so that the emir Abd al-Rahman II could listen to three young *medina* singers (Muḥtālah, Maḥāriq, and Mu'allilah) who had just entered the Alcázar (Ibn Ḥayyān 2001, *Muqtabis* II,1 [153v], p. 212 from the trans.). Two slaves with the same origin (Faḍl and her companion 'Alam) "*gave name to the "Medina house" in the Alcázar*" (Ibn Ḥayyān 2001, *Muqtabis* II,1, [147r], p. 192 of the trans.). Other rooms probably built by 'Abd al-Raḥmān II himself may have held similar activities such as Dar al-Surūr (The Pavilion of Happiness) (Ibid. [164v], p. 247 of the trans.) or the "House of the Pebbles" (Ibid. [162v], p. 242 of trans. and [163v], p. 245 from trans).

It is highly likely that some of these spaces or buildings had a multi-functional nature, as occurs in the first constructive phases of Madīnat al-Zaḥrā, where "*the ceremonious functions are developed under the same models as the residential ones, without knowing of architectural types to differentiate one from the other*" (Vallejo Triano 2010, p. 484).

It is not currently possible to apply the principle by which an architectural form is identified with a function in the Alcázar of Córdoba. The few structures documented up until now inside the Alcázar (except for the *hammam*) do not allow for the definition of the specific activities they were destined for. In any case, the elements around which the excavated rooms are distributed are paved courtyards with calcarenite or marble slabs (such as the Women's Courtyard) (León-Muñoz 2020a, pp. 293–98), or with lime mortar painted in red (The Bishop's Gardens) (Castejón y Martínez de Arizala 1961–62b), in this case, associated with square pilasters. These pavements could have formed part of the portals that preceded the reception rooms or corresponded to the courtyards of buildings with a residential nature. There is no sign of spaces recorded that aimed at official functions and representation, such as the basilica-floor rooms that are so characteristic in Madīnat al-Zaḥrā, introduced from the extensive renovations performed in the 950s (Vallejo Triano 2010, p. 490). This does not invalidate the possibility that the Umayyad Alcázar of Córdoba may serve as an example or architectural model for the buildings of Madīnat al-Zaḥrā. The knowledge level of both enclaves is quite uneven, and the scarce surface excavated of the palatial complex of Córdoba impedes, for the time being,

establishing any conclusion on its type. On the other hand, as we have indicated, the distribution and nature of the buildings of the Al-Andalus palace complex was highly conditioned by the existence of preceding structures to which they adapted, reused, and included in the new buildings.

Although the information available is indirect, we have some signs that allow us to suggest a hypothesis of the identification of an oratory or a small mosque associated with the Alcázar. In a map raised in the middle of the 17th century (1662), corresponding to the archives of the Inquisition (Cuadro García 2004), a structure located outside the Alcázar de los Reyes Cristianos, alongside the round tower of the south-western angle (Tower of the Inquisition) is highlighted (Figure 15). It is a rectangular-shaped floor building, with a NE–SW direction, which is described in the key to the map as: "Chapel of San Acacio, a very good set and strong work". In a document from 1578, the "Royal Chapel of these Alcázars" is mentioned alongside the "Cube Tower" (Gracia Boix 1981, p. 113).

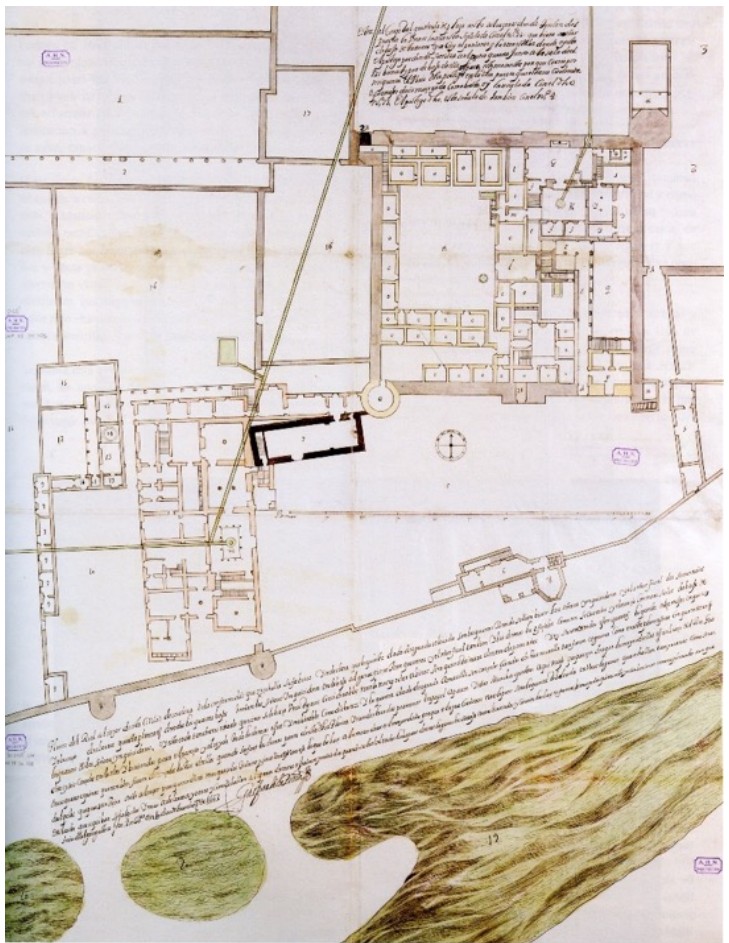

Plano del Alcázar de Córdoba, 1662, Ministerio de Cultura, Archivo Histórico Nacional, Madrid, Sección Inquisición, MPD, nº 426, Leg. 2425.

**Figure 15.** Map raised in the middle of the 17th century (1662) corresponding to the archives of the Inquisition. It is a rectangular-shaped floor building, with a NE–SW direction, which is described in the key to the map as: "Chapel of San Acacio, a very good set and strong work", probably reusing a previous mosque (Cuadro García 2004).

The particular setting and direction of the structures, which is wholly divergent from the rest of the complex, and its use as a place of worship after the Castilian conquest allows us to hypothesise that it may have involved the reuse of a building originating from a similar religious function. The type of floor does not correspond to the previously established canons of the urban Córdoba mosques, with a more or less square-shaped prayer room, a courtyard, and minaret. This would be, due to its location and size, a private

oratory or one with restricted access, which would explain its relatively rare presence in written Al-Andalus sources. In this sense, it would be possible to suggest its possible identification with one of the mosques raised at the foot of the Alcázar by Hišām I in 793, which is reported by Al-Maqqari (in the version by Pascual de Gallangos): "*In the days of Hišām, the city of Narbonne was newly taken from the Christians. After Gallaecian vassals requested peace, Hišām only granted it on these harsh conditions, one of which was to transport a certain number of loads of soil from outside the demolished walls of the conquered city, Narbonne, to the door of his own palace in Córdoba, to be used here for the construction of a mosque, facing the Gate of the Gardens; and not only building the mosque with these materials, but a great amount still remained in front of the royal palace*" (Gayangos 1840, History, vol. I, p. 99). In another passage, al-Maqqari, using the information of Ibn Baškuwāl, indicates that in front of the two southern doors of the Alcázar (Bāb al-Sudda and Bāb al-Ŷinān) "*the road that rises over the Guadalquivir is found, in which two mosques, famed for their holiness, are found, where the emir al-Ḥakam I administered justice to the oppressed, hoping to achieve Allah's reward*" (Rubiera Mata [1981] 1988, pp. 122–23). Another option, though less likely in our opinion, is that it could be identified with an oratory from the Caliphate era, perhaps associated with one of the capitals studied by M. Ocaña, in which it reads "Work of Safar for the oratory of his lord" (Ocaña Jiménez 1935, pp. 161–62; Torres Balbás 1957, p. 591).

Definitively, the Córdoba Alcázar must have been organised around a complex and dense network of reception rooms, residential, and service rooms that extended without continuous solution and occupied all the space available inside the defined area, as we have proposed, in the era of 'Abd al-Raḥmān II. Although it is not possible to identify any of these spaces from the archaeological information, we have tried to highlight in which of them the power of the dynasty is manifested. And not only in the Caliphate period, but in a diachronic process during several centuries of the history of al-Andalus, in which the occupation of the same space by successive rulers is one of the main arguments to legitimize their power.

### 6. Conclusions, Assessment, and Future Perspectives

After this brief tour through the information currently available regarding the Al-Andalus Alcázar of Córdoba, we may deduce some conclusions:

The Alcázar of Córdoba is, together with the Congregational Mosque, one of the pillars of the Umayyad dynastic legitimacy. Similarly, a functional continuity phenomenon regarding the preceding urban spaces is produced: in the case of the mosque, in the episcopal group of the Late Antiquity city and in the palace, making the most of the civil complex of the city.

Inside, acts with profound symbolic and propaganda value such as the claim of sovereignty in the throne room and burial alongside one's predecessors were performed. On its doors, particularly in Bāb al-Sudda, they appear before their subjects and their authority is made evident.

The palatial complex is the result of a complex constructive process that took various centuries, linked to historical events and the political projects of the governors of the Umayyad dynasty. Therefore, for example, the intense architectural remodelling program undertaken by 'Abd al-Raḥmān II is a clear reflection of the deep transformation that was performed in the early Andalusian Arabian Umayyad State.

The Alcázar that 'Abd al-Raḥmān I took possession of, or the area in which al-Ḥakam I hid during the revolt of the people of the southern suburbs, were by no means the same as the palace where the receptions that Al-Ḥakam II presided over at the end of his reign took place. For this reason, the analysis of this architectural complex can only be addressed from a diachronic perspective, given that the successive renovations and expansions not only modified its perimeter (as occurred in a very evident manner in the Almohad era), but that they also substantially changed the internal layout.

The rich and varied textual information, which makes it difficult to assimilate details from other studies, may not condition the proceedings of the research, which should be

based mainly on the material evidence. Additionally, the gaps and a complete translation of the information on the Alcázar should still be addressed, as indicated by María Jesús Viguera: "*The textual references are pieces of a puzzle that we still have not completed in whole, but also in relation to the Alcázar of Córdoba they need to establish all the passages that are found in the Arabian sources, with their careful and homogeneous translation*" ([Viguera Molins 2020](#), p. 134).

The material data recovered up until now is insufficient given the limited extension of the excavated areas and does not allow us to identify the internal spaces or rooms of the Al-Andalus Alcázar. However, the scarce records available allow us to obtain a general characterisation of the complex between the 5th and 13th centuries. This analysis should not be exclusively restricted to the inside of the walled area, but also include its immediate surroundings, as the buildings in this urban sector must have been closely linked through their functionality with the Alcázar itself (promenade, *al-Raṣīf*, Albolafia, pavements, and walls of a monumental nature, etc.) or with the mosque (sābāt).

Only the continuity of the research and its integration into a general and systematic program that contemplates and plans the archaeological activities, architectural interventions, and activities of dissemination and heritage recovery will allow for the numerous gaps presented by this extensive and complicated architectural complex, seat of the power of Al-Andalus for various centuries of its history, to be filled in the coming years.

**Funding:** This research has been carried out in the framework of the R+D+i Project "From Iulius Caesar to the Catholic Monarchs: Archaeological Analysis of 1500 Years of History in the Mosque-Cathedral of Cordoba and its Urban Surroundings" (DE IURE, Ref.: PID2020-117643GB-I00), granted by the Science and Innovation Ministry, belonging to the State Programs of Generation of Knowledge and Scientific and Technological Strengthening of the R&D&i system I+D+i system, in its 2020 call.

**Data Availability Statement:** The data presented in this research have been published in local magazines, conveniently cited, or are the result of archaeological interventions directed by the author or by colleagues within the framework of the Collaboration Agreement between the Municipal Urban Planning Bureau (GMU) and the University of Córdoba (UCO). The most recent data has been provided by colleagues who have given their authorization for the use of said information and images. The original and unpublished excavation reports used in this work are available at the Delegación de Cultura de Córdoba, c/Capitulares, 2, 14071, Junta de Andalucía, Córdoba (Spain). Phone: +34957015300, e-mail: dtcordoba.ctcd@juntadeandalucia.es.

**Conflicts of Interest:** The author declares no conflict of interest.

## Notes

[1] We refer to this exhaustive work for the aspects related to the archaeological activities and discoveries produced up to the end of the 20th century ([Montejo Córdoba and Garriguet Mata 1998](#), pp. 309–14).

[2] This work proposes, reproducing the hypothesis of Murillo ([Murillo Redondo and León-Muñoz 2019](#)), the location of the main buildings that are part of the complex and mentioned by the sources. Regarding the archaeological information, a date from the 10th century is attributed for the stretches of wall currently preserved: "The elements of the wall preserved today date to the tenth century" ([Arnold 2017](#), p. 21). This statement should be nuanced as it assigns a generic chronology to the whole complex from a part of the northern section in the "Parking la Mezquita", where the preserved date of the eastern parts, as we shall see, is from the 9th century.

[3] "*The Prince of the Believers went at the front, in the direction of the Musāra, in the western end of Córdoba. There he was received by some of the most important members of the Qurayš and a group of māwlas, who stepped on the ground, blessed and worshiped him. Afterwards, they continued to the great market of Córdoba (...). From there, always embraced by successive groups of upper and lower people, he followed his route to the Alcázar of Córdoba, which he entered through the Iron Gate, located in the midday, in an unrivalled raid*" ([García Gómez 1967](#), p. 253).

[4] In the case of Toledo, recent hypotheses place the court area complex in the Vega Baja, in the suburban area, and they include as part of this complex the Basilica of Saints Peter and Paul (*ecclesia praetoriensis sanctorum Petri et Pauli*) ([Teja and Acerbi 2010](#), p. 4).

[5] The Islamic sources mention in the moment of the conquest, the existence in the southern zone of the city of various splendid residences (called bālat-s) belonging to the Visigoth aristocracy. Perhaps these are some of "*the building surrounding*" [the palace], mentioned in the afore cited al-Maqqari.

6    We do not know whether the structures documented in 1928 were part of the same area: "*in work for the sewers and beside the tower of the Alcázar by the Ronda de Isasa, remains have been discovered from the lesser part of a Roman wall forming an angle, one of whose paraments went to the north and another towards the interior of the Alcázar*" (Santos Gener 1928, p. 21).

7    The archaeological material recovered from this layer of red clay dates from the first half of the 9th century.

8    This bath may be the one mentioned by Ibn Iḍārī in the story of the death of Al-Mustaẓhir bi-llāh Abū al-Muṭarrif ʿAbd ar-Raḥmān (V) in 1024, who unsuccessfully tried to escape from the Alcázar through the bathroom door, into which an oven he was located and stuck, to finally die on the order of Muhammad III (Ibn Iḍārī 1993, *La caída del Califato*, [138–139], pp. 122–23 from the trans.).

9    Huici translates: "*The day after the already-mentioned sacrifice party, he sat at dawn in* his seat, happy *for his Alcázar of Córdoba, for the audience of greetings and congratulations…*" (Ibn Ṣāḥib al-Ṣalāt 1969, *Al-Mann bi l-imāma*, p. 185 from translation).

10   During the brief period of just eight months in which Córdoba was the government seat of Al-Andalus in 1162, with Abū Yaʿqub—future caliph—and Abū Saʿīd, until their definitive transfer to Seville, the Almohad chronicler Ibn Ṣāḥib al-Ṣalāt reports that "*they ordered to build their palaces and other buildings and to fortify their borders, and they brought builders, architects and workers to build their palaces and the houses in the neighbourhoods so they could build again. Their state was built and improved. The architect Aḥmad b. Bāso took care of it, who repaired all that which had been destroyed…*" (Ibn Ṣāḥib al-Ṣalāt 1969, *Al-Mann bi l-imāma*, [206], p. 50 de trad.).

11   The excavations mentioned are as follows: C/Terrones, 4 and 6, C/Postrera, 5, C/Enmedio, 2, and 12, and the covered pen in the Royal Stables (Montejo Córdoba 2015, p. 5).

12   The names of some of these palaces are as follows: al-Kāmil (The Perfect), al-Muŷyaddad (The Renewed), al-Ḥaʾir (The Dam); al-Rawḍa (The Garden); al-Zāhir (The Shining); al-Maʿšūq (The Beloved), al-Mubārak (The Blessed), al-Rašīq (The Elegant), Qaṣr al-Surūr (The palace of Joy), al-Taŷ (The Crown), and al-Badīʿ (The Marvelous) (Al-Maqqari, *Nafḥ al-Ṭib* II, pp. 11–13, taken from Rubiera Mata [1981] 1988, p. 122 and from Arjona Castro 1982, p. 207, doc no. 273).

13   The exception to the rule is the death of the caliph al-Mustaẓhir bi-llāh (ʿAbd al-Raḥmān V), assassinated during the *fitna* (in 1124) in the boilers of the bath of the Alcázar, decapitated by al-Mustakfī (Muḥammad III) by his own hand, after which "*he was taken to his house and buried there*" (Ibn Iḍārī 1993, *La caída del Califato* [64] p. 219 from the trans.).

14   García Gómez indicated so in similar terms: "*A royal palace (…) is already within itself a symbol of royalty: to occupy it is to invest oneself in its power. Its prestige comes from ºits antiquity*" (García Gómez 1965, p. 320).

15   ʿAbd al-Raḥmān II built a pylon with the excess water, "*before south central [the door] of the Alcázar, the called Garden Gate (Bāb alŷinān), where it poured into a pile of marble which all the people who went to the Alcázar or passed by it had access to*" (Ibn Ḥayyān 2001, *al-Muqtabis* II, 1 [140r], p. 172 from the trans.). ʿAbd al-Raḥmān III "*also ordered that a pylon be built in the fountain of the sewage who was at the entry of the Alcázar and its door, called the Gate of Lattice*" (*An anonymous chronicle*, [28] p. 126 from trans.).

16   Marfil Ruiz (2010, pp. 464–65) proposed, from the structures documented in the 2008 intervention, the location of this door in the southern courtyard of the Episcopal Palace. However, the excavation led by R. Ortiz in 2015 obligates us to fully discard this hypothesis, as that interpreted as one of the buttresses of the door is, in fact, part of the south parament of a strong retaining wall towards the west.

17   "(…) also «*he made (sic) the terrace that dominates the main gate… of the caliph's Alcázar, the first southern one, called Gate of the Zuda (Bāb al-sudda), placing it on top like a crown, which sealed its extraordinary grandeur*»" (Ibn Ḥayyān 2001, *Muqtabis* II-1, p. 172).

18   The express mention of the Western Room (which led García-Gómez—1965—to suppose the existence of an eastern one) is repeated at Madinat al-Zahra. This duality has been subject to a recent revision of the Caliph city, from which the identification of the Western Maŷlis with the throne room of the sovereign is proposed, while the eastern room, linked to the heir, would correspond to the Central Pavilion (Vallejo Triano 2016, p. 454).

19   Also called *Turbat al-Julafāʾ* (*Ḏikr balād al-Andalus* 1983, p. 142 of trans.).

20   According to this hypothesis, the Rawḍa would be found "*in the area of the current San Basilio street in the Alcázar Viejo neighbourhood, as has been verified through the various work that has destroyed its remains*" (Marfil Ruiz 2004, p. 57). However, there is no record of any material remains linked to this funeral space.

21   Even if the visible remains of its use as a waterwheel has been dated from the Christian Late Middle Ages (Córdoba de la Llave 2020), the existence of a waterwheel is recorded in the stamps of the city in the 14th century and an Almoravid era date has been proposed (1136–1137) from an unedited text located by Lévi-Provençal (Torres Balbás 1942, p. 462). However, its origin may be located in the 9th century from some documentary and architectural signs (Torres Balbás 1940, pp. 204–5; Ocaña Jiménez 1975, p. 40). Only an exhaustive archaeological analysis of the preserved structures will allow for the clarification of any doubts that are still caused by this unique building.

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
