# Peer review of "The Alcázar of Córdoba: The Seat of Islamic Power in Al-Andalus"

_arts, 2023_

Round 1

Reviewer 1 Report

The paper provides a comprehensive, up-to-date summary of the archaelogical investigation in the Alcázar of Córdoba. Although a similar summary has already been published in Spanish, the paper is very useful for an international readership. Particularly interesting are some issues raised in the final section, for example the importance of gates, throne halls and tombs for establishing legitimacy. Only a few interpretations proposed in the paper might need a more in-depth discussion, for example why the rows of columns of the visigothic period cannot be interpreted as a columned street, how certain the western limit of the alcázar is (a map whould be useful), and how sure the identification of the mosque within the alcázar is. Fig. 1 and 2 are missing.

The English language should be checked by a native speaker. Some words are not used correctly or not in the usual way ("fan", "project"/"proposal", "bibliographic production", "news", "research vocation", "intervention trails", "septentrional" instead of "northern", "of all and its parts" etc.). Some translations are not correct (caliza = "limestone", not "chalk", soga = "stretcher", not "rope",  autor = "author", not "scribe"). Some names and terms should better not be translated, to avoid confusion ("Royal Academy of Córdoba", "courtyard of the Oranges", "door of the Viziers", "Preventive Archaeological Activity", "Tolosa" = Toulouse). Please adapt the Arabic transliteration from the Spanish to an international system (à , Å·, j, á should not be used in this way). Please reconsider how to translate the chronological terms “emiral” and “califal” (for an international readership "early Emir chronology", "emir era", "Emir stage", "Emir renovations", "emir courtyard", "caliph era", "caliphate activities", "caliphate hammam" do not make sense). Maybe replace by “8th/9th century” and “10th century”, or by “(period) of the Emirate” and “(period) of the Caliphate.”

Author Response

Dear Sirs:

First of all, I sincerely appreciate the review of the manuscript and, especially, the corrections and comments made by the specialists. I must apologize for the errors made in the translation, which have been corrected thanks to reading and proofreading by a native English speaker.

Some of the "signposts" have also been included at the indicated points to clarify or synthesize the fundamental idea of each chapter. In other cases, I consider that a more extensive explanation would make the text excessively long and could be repetitive.

Issues related to the layout (indentations, text format or absence of images 1 and 2) correspond to the editor and I understand that they will be corrected in the final layout.

Best regards.

Reviewer 2 Report

This article is wonderfully rich in new archaeological information about the Alcazar of Cordoba, a palace complex that up until recently has been little studied and little understood. The author presents a compilation of archaeological discoveries in historical layers to demonstrate that the complex was built over time and that Emirs retained certain historical features to assert royal legitimacy.

While I find the arguments compelling and interesting, I think this article would benefit from considerably more signposts and a thorough language edit. Many passages require greater clarity. My comments below reflect a few questions that I had as I read and areas in need of expanding/refining in terms of reminding the reader about the overall thesis throughout (signposts).

Abstract - the accent is missing on Cordoba. At the end of the abstract, the author should explain in a sentence why this information will be important to the reader. What will be learned that is new? I suggest that here it would be good to explain that the reoccupation/appropriation was a political choice. That sort of gets buried in the last section of the article.

Normally I see line numbers to help me pinpoint where to edit, but I don't see that here, so I'll just do my best to highlight some areas.

Paragraph 1 - "to which it was attached" ;  "This lack, which is substantially more evident..." than what? 

Paragraph 2 - "Another factor to bear in mind" - for what purpose? about the lack of publications?

p. 2 para 1 - "complex: on the one hand, the difficulty when it comes..." awkward - refine

p. 2 para 2, end - a new reading of space is suggested, but this should be explained further. What new point of view will be uncovered? Why does the reader want to read the rest of this article? What new ideas will surface because of this study? Lead the reader with some signposts. 

Section 2, para 2 - "Therefore, the work aimed..." work can't "dedicate", only people can. The use of the word "news" in this section and elsewhere should be reevaluated. Chronicles? Reports? It seems anachronistic.  "establish a general frame" (what kind of frame, architectural? spatial? chronological) unclear. At the end of this paragraph, we need a sentence that explains why the reader should read this state of research section in light of your thesis. What will all this lead to? Why should the reader care? Signpost

p. 3, para 1 "in which he orderly copies"?? 

p. 3, para 3 - tenses switch here

p. 5, para 1 - "The new info obtained from this excavation allowed (scholars?) to."  "As for the main news"?? Overuse of the word "which" in this sentence, too. Be careful of overusing "which" throughout the article. "On the other hand, the results allow to specify" = refine this sentence." Buried here at the end of this paragraph appears to be your thesis. Here I ask - so, is this what scholars began to do, or is this diachronic approach totally new and uniquely your idea? Make your contribution more obvious. Otherwise it seems you are just chronicling what has been done by others without purpose.

p. 5 last paragraph, "The integration of this data has allowed (me? other scholars in general?) to suggest..." Unclear

p.6 - because I don't have any images up to this point, I began to get a little lost by this page. Hopefully your plans will clarify things.

p. 6 para/ 2 - "In order to do this" (do what? what is "it" referring to here"?

p. 6 para. 4 - Can you hint at this information at the outset of your section here? Signpost this.

3. The Precedents, para. 1 - this first sentence is not a great topic sentence and it seems to me it belongs at the beginning of the second paragraph. 

p. 7 para 3 "in a moment to still assess better" ?

p, 7 para 3 italicize "c" in cardo maximus; issue of "news"

The paragraph justifications and the indented quotations begin to be off following this page. I got confused about what was a quotation and what was your analysis.

be consistent with capitalization of direction words (Western facade, southern closure).

p. 10, para 1 - "As such, we have a clear..." missing something here.

p. 10, para 2 - "From the archaeological info available..." hint at this in the intro to signpost your thesis intentions. 

p 11, para 3 - This paragraph and then next both begin with "Regarding" Vary. Also the next paragraph needs indentation. Check paragraph indentation throughout as I see this again on p. 13

p. 14, para 2, "In the Women's Courtyard..."this "emir renovations" seems strange to me. Maybe "renovations from the emir era"?

p. 15, para 2 ""According to that stated" is awkward in English.

p. 15, para 3, first line - check tenses

p. 16 - the justification needs left alignment. "In the Women's Courtyard...overuse of which in this sentence

p. 18 - end of section - we need a recap of how all this relates to your thesis. I'm getting a little lost about the main idea by this point. 

p. 22 - "Definitively, the look...should be far..." refine this sentence

p. 22, para 3, first sentence - unclear

p. 24, section 5.2, first para - at the end - So, what areas of architecture will be addressed in this section and why will the reader care? Signpost thesis.

p. 25, para 1 - Can you explain why this will matter? Maybe mention the main elements of this section, like the door and the throne? Order them because I get confused in this section. How is this level distinguished from earlier levels? 

Why do doors "matter most"? Is this true of the earlier timeframe? This idea about the doors is very interesting, but could be expanded in terms of political meaning. I think this section is very rich in terms of your overall contribution. Highlight that!

p. 28, para 2 - "of which the Umayyad sovereigns were very fond."

Before you get to the Conclusion section, you should go back and add clear signposts at the beginning and end of each of the prior sections. Right now, the sections read as a laundry list of discoveries, and the reader has little idea about what you are contributing that is new. Build your analyses so that your conclusion has more power at the end. Your analysis of each level is buried in each section and should be highlighted more strongly in the concluding paragraph of each section to help remind the reader about the significance of the architectural layers. 

For example, on p 30, para 2, "Definitively..." my question is "so what?" 

p. 31 - line 1 - unclear sentence

The language needs refining in several passages.

Author Response

(The authors gave the same response as above.)
